# Disrupting cellular memory to overcome drug resistance

Guillaume Harmange [1], Raúl A. Reyes Hueros[2], Dylan L. Schaff[3], Benjamin Emert[4], Michael Saint-Antoine[5], Laura C. Kim [6], Zijian Niu[7,8], Shivani Nellore[9,10], Mitchell E. Fane [11], Gretchen M. Alicea[12], Ashani T. Weeraratna[12,13], M. Celeste Simon [6,14], Abhyudai Singh [5] & Sydney M. Shaffer [3,15]

Gene expression states persist for varying lengths of time at the single-cell level, a phenomenon known as gene expression memory. When cells switch states, losing memory of their prior state, this transition can occur in the absence of genetic changes. However, we lack robust methods to find regulators of memory or track state switching. Here, we develop a lineage tracing-based technique to quantify memory and identify cells that switch states. Applied to melanoma cells without therapy, we quantify long-lived fluctuations in gene expression that are predictive of later resistance to targeted therapy. We also identify the PI3K and TGF-β pathways as state switching modulators. We propose a pretreatment model, first applying a PI3K inhibitor to modulate gene expression states, then applying targeted therapy, which leads to less resistance than targeted therapy alone. Together, we present a method for finding modulators of gene expression memory and their associated cell fates.

Gene expression memory describes the length of time that a particular expression state exists in an individual cell or lineage of cells. Gene expression memory can exist over a range of different timescales and is ultimately a quantitative measurement[1,2]. In cancer, an intermediate timescale of memory has been associated with several important phenotypes including, stemness, differentiation[3,4], metastasis[5], and drug resistance[2,6–8]. In these examples, the cellular state underlying the phenotype persists through multiple cell divisions, but is ultimately not permanent, and thus amenable to switching states. For cancer therapy resistance, drug-naive melanoma cells can seemingly randomly fluctuate between two states, one which is susceptible to targeted therapy, termed drug-susceptible, and another which would become resistant if the drug is applied, termed primed for drug resistance[9,10]. This finding

[1]Cellular and Molecular Biology Graduate Group, Perelman School of Medicine, University of Pennsylvania, Philadelphia, PA, USA. [2]Department of Biochemistry and Molecular Biophysics, Perelman School of Medicine, University of Pennsylvania, Philadelphia, PA, USA. [3]Department of Bioengineering, School of Engineering and Applied Sciences, University of Pennsylvania, Philadelphia, PA, USA. [4]Division of Biology and Biological Engineering, California Institute of Technology, Pasadena, CA 91125, USA. [5]Department of Electrical and Computer Engineering, University of Delaware, Newark, DE 19716, USA. [6]Abramson Family Cancer Research Institute, Perelman School of Medicine, University of Pennsylvania, Philadelphia, PA, USA. [7]Department of Chemistry, College of the Arts and Sciences, University of Pennsylvania, Philadelphia, PA, USA. [8]Department of Physics, College of the Arts and Sciences, University of Pennsylvania, Philadelphia, PA, USA. [9]Department of Biology, College of the Arts and Sciences, University of Pennsylvania, Philadelphia, PA, USA. [10]The Wharton School, University of Pennsylvania, Philadelphia, PA, USA. [11]Cancer Signaling and Microenvironment Research Program, Fox Chase Cancer Center, Philadelphia, PA, USA. [12]Department of Biochemistry and Molecular Biology, Johns Hopkins School of Public Health, Baltimore, MD, USA. [13]Sidney Kimmel Cancer Center, Johns Hopkins School of Medicine, Baltimore, MD, USA. [14]Department of Cell and Developmental Biology, Perelman School of Medicine, University of Pennsylvania, Philadelphia, PA, USA. [15]Department of Pathology, Perelman School of Medicine, University of Pennsylvania, Philadelphia, PA, USA. ✉e-mail: sydshaffer@gmail.com

highlights the ability of clonal cancer cells to transition between different stable phenotypes.

Because drug resistance can emerge from cells that are in a primed gene expression state, one intriguing possibility to prevent resistance is to transform primed cells into drug-susceptible cells. However, we currently do not know the molecular cues that trigger cells to switch between these states. Knowing these cues would make it possible to therapeutically target state-switching pathways to drive cells out of the primed gene expression state and sensitize them to therapy. Such an approach requires deep characterization of the processes underlying state switching, which remains exceedingly difficult with available methods.

Currently, there are limited techniques that can reveal memory and state switching in single cells. Our previous work inferred cellular memory from bulk RNA-seq measurements[2], but it failed to capture drivers of switching between memory states. On the other hand, single-cell RNA-sequencing (scRNA-seq) can capture the heterogeneity of a population, but it fails to capture the timescales for which different states have been present in individual cells. Several computational and experimental techniques have been developed to resolve time in single cells on short timescales, on the order of hours[11,12], but few exist for the longer timescales of days to weeks, as needed to track gene expression memory. Recent advances in high-throughput cellular barcoding technologies have made it possible to track cellular lineages across any length of time[13]. Pairing cellular barcoding with scRNA-seq enables us to now match cellular lineages with their transcriptome[8,10,14,15] and is thus an ideal tool for tracking transcriptional states across lineages to measure cellular memory.

Here, we present a method for measuring memory by combining cell barcoding and scRNA-seq called scMemorySeq. Our experimental design uses a controlled number of cell divisions to capture lineages that have undergone state switching. We apply scMemorySeq to drug-naive human melanoma cells and find two distinct gene expression states, which correspond to drug-susceptible and primed cell populations. Through lineage tracing with cell barcoding, we identify cells that switched between states and determine that the TGF-β and PI3K pathways control state switching. Ultimately, we find that by initially disrupting the primed state through PI3K inhibition and then applying a BRAF inhibitor in combination with a MEK inhibitor (BRAFi/MEKi), we can reduce the frequency of drug resistance. Taken together, we demonstrate the feasibility of molecularly targeting memory and state switching to eliminate gene expression states in cancer that prime cells for undesirable phenotypes.

## Results

We first sought to identify the molecular pathways that underlie cellular memory in drug-susceptible and primed cells in melanoma (Fig. 1A). Based on our previous work, we know that both the drug-susceptible and primed states exist in untreated melanoma cells and that cells can fluctuate between these states[2]. When targeted therapy is applied, cells in the primed state have a higher likelihood of resistance, whereas cells in the drug-susceptible state succumb to the treatment[9]. To identify molecular regulators of switching between these two states, we developed a technique called scMemorySeq that identifies heritable gene expression states and state switching by combining cellular barcoding and scRNA-seq. In our experimental design, the cellular barcodes enable high-throughput tracking of cells over any desired period of time while the scRNA-seq reveals the transcriptional states of every cell. Ultimately, we infer cellular memory by examining the gene expression states of cells within the same lineage. States that have memory over the experimental timescale result in lineages in which all the cells at the end state have the same gene expression state as the initial cell (Fig. 1B). However, when memory is lost over the experimental timescale, then the end state consists of cells with multiple gene expression states, some of which are different from the initial state (Fig. 1B).

We applied scMemorySeq to BRAF V600E mutated WM989 melanoma cells containing the drug-susceptible and primed states[2,9,10]. We transduced drug-naive cells with a high-complexity viral barcode library consisting of a transcribed 100 base pair semi-random barcode sequence in the 3' UTR of GFP[10] (Fig. 1C). To determine the initial state of cells, we sorted the primed cells using known primed cell markers (EGFR and NGFR) and also sorted a mixed control. Because the primed state is rare, representing ~2% of the population, the mixed sample predominantly consists of drug-susceptible cells. We then allowed the cells to expand through roughly 4 doublings (12–14 days). Over this time, we expect that most cells will maintain the memory of their initial gene expression state, but that a subset of cells will lose the memory of their initial state, thereby capturing state-switching events. At the endpoint, we harvested these cells for scRNA-seq and performed a PCR side reaction to specifically amplify the lineage barcodes from each cell.

We profiled a total of 12,531 melanoma cells (7581 cells with barcodes) and found two major transcriptionally distinct populations (Fig. 1D, Supplementary Fig. 1A, C). One of these populations expressed genes associated with the primed state, including EGFR and AXL[2,9,10], and the majority of the sorted NGFR and EGFR-high cells were included in this cluster (Fig. 1D, Supplementary Fig. 1B). The other population showed higher expression of genes associated with the drug-susceptible state, including SOX10 and MITF (Fig. 1D, Supplementary Fig. 1D). The distinction between these two states is robust and seen in high dimensional space by Louvain clustering and by multiple dimensionality reduction methods (Supplementary Fig. 1A, C, E–H).

Within the cluster of suspected primed state cells, we observed significant heterogeneity in previously described primed state marker genes, including EGFR, AXL, and NGFR, with each gene expressed by some of the cells in the cluster, but not all of the cells (Fig. 1D, Supplementary Fig. 1D). Furthermore, in a separate experiment, we profiled the chromatin accessibility of EGFR and NGFR-high cells and found epigenetic differences between these populations further confirming that there is additional heterogeneity within the primed state (Supplementary Fig. 1I). This observation suggests that the individual primed cell markers from previous work[9] only captured a subset of primed cells. We identified a different marker, NT5E, that encapsulated the entire cluster containing the primed state population (Fig. 1D). To determine if NT5E is an effective marker of primed cells, we stained WM989 cells with NT5E antibody and sorted the top 2% of cells. We applied the targeted BRAF inhibitor, vemurafenib, and found that the NT5E-high samples had 5.5-fold more resistant cells compared to NT5E-low cells (Fig. 1E, Supplementary Fig. 2A, B). Thus, we demonstrated that the marker gene NT5E captures the entire cluster of transcriptionally similar cells and that these cells are indeed more likely to be resistant to targeted therapy. We also found that primed cells have similarities to previously published gene expression states associated with drug resistance in melanoma (Supplementary Fig. 2C)[16–24]. Therefore, we concluded that cells in the NT5E-high cluster are in a gene expression state that primes them for drug resistance, while cells in the other cluster are in a drug-susceptible gene expression state (Fig. 1F).

With a defined gene expression state for primed cells, we next assessed the generalizability of the primed state in tumor models and patient samples. We used RNA FISH HCRv3.0 on tumor samples derived from WM989 cells grown in NOD/SCID mice[6,25]. We used probes for NT5E and SOX10 on tumor sections and quantified expression across 5,600 cells. We found rare cells expressing high levels of NT5E mRNA scattered throughout the tissue (Fig. 2A, additional images in Supplementary Fig. 3A). SOX10 showed diffuse expression across the tissue, but many of the NT5E-high cells did not have SOX10 expression, as predicted by the scRNA-seq. Altogether

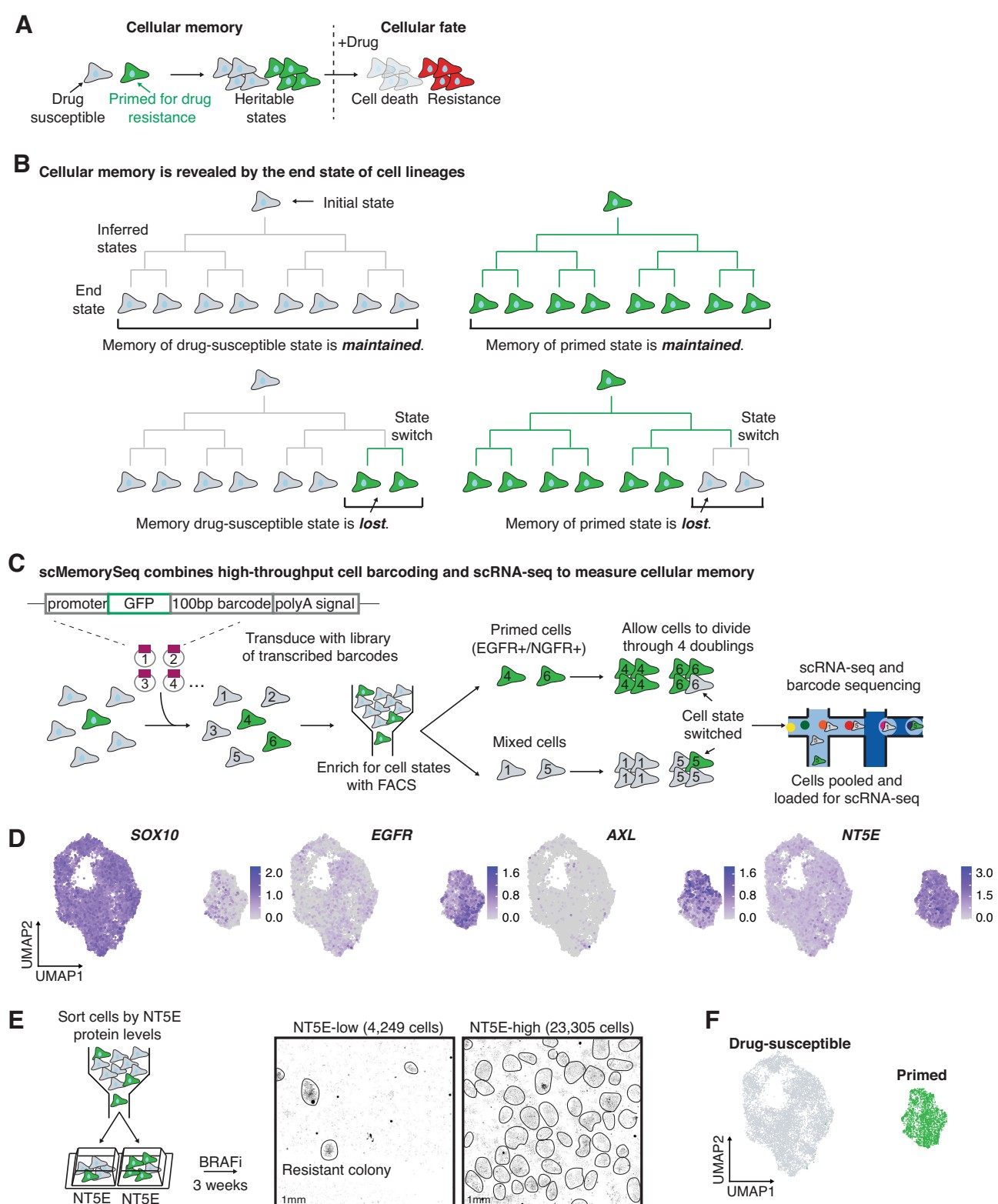

**A** Cellular memory | Cellular fate

Drug susceptible → Primed for drug resistance → Heritable states → +Drug → Cell death, Resistance

**B** Cellular memory is revealed by the end state of cell lineages

← Initial state
Inferred states
End state

Memory of drug-susceptible state is *maintained*.

Memory of primed state is *maintained*.

State switch
Memory drug-susceptible state is *lost*.

State switch
Memory of primed state is *lost*.

**C** scMemorySeq combines high-throughput cell barcoding and scRNA-seq to measure cellular memory

promoter | GFP | 100bp barcode | polyA signal

Transduce with library of transcribed barcodes

Enrich for cell states with FACS

Primed cells (EGFR+/NGFR+)

Allow cells to divide through 4 doublings

Mixed cells

Cell state switched

scRNA-seq and barcode sequencing

Cells pooled and loaded for scRNA-seq

**D** *SOX10* *EGFR* *AXL* *NT5E*

UMAP2 / UMAP1

**E** Sort cells by NT5E protein levels

NT5E low / NT5E high

BRAFi 3 weeks

NT5E-low (4,249 cells) — Resistant colony — 1mm

NT5E-high (23,305 cells) — 1mm

**F** Drug-susceptible / Primed

UMAP2 / UMAP1

these *NT5E*-high/*SOX10*-low cells demonstrate that the primed state exists both in vitro and in vivo (and is not an artifact of cell culture conditions). We also performed scRNA-seq on a different melanoma cell line, WM983B, and found a subpopulation of cells with a large number of primed state markers, including *NT5E* (Fig. 2B, C). To establish whether the primed states exist in patient tumors, we analyzed scRNA-seq data[26–28], which included 7 samples directly from human tumors and found that 5 of them had a subpopulation of cells

with high expression of genes associated with the primed state (Fig. 2D, E, Supplementary Fig. 3B). Furthermore, a previous analysis of data from the cancer genome atlas showed that the presence of *MITF-low/AXL-high* cells (as also seen in our primed cell state) in drug-naive patient tumors was predictive of a shorter progression-free survival rate[29]. Together, these data demonstrate the generalizability of the primed cell state and suggest that it might be predictive of response to BRAFi/MEKi.

**Fig. 1 | scMemoryseq can reveal when cells change state. A** Model for how the heritable primed state leads to resistance to targeted therapy. Cells primed for drug resistance (in green) proliferate and pass on their gene expression state through cell division, which demonstrates the concept of gene expression memory. These primed cells survive treatment with BRAFi and MEKi (resistant cells in red) while the drug-susceptible cells (in gray) die. **B** Schematic of lineages that maintain memory compared to those that lose memory. In lineages that maintain memory, the end state of all the cells is the same as the initial state. In lineages where memory is lost, the end state contains a mixture of cell states. **C** Schematic of the experimental design we used to capture the transcriptome and lineage of cells. We transduced melanoma cells (WM989) with a high-complexity library of lentiviral lineage

barcodes. We then sorted a sample of primed cells (based on EGFR and NGFR) and a mixed population. This sorting step provides us with the initial state of the cells. We then allowed the cells to undergo approximately four doublings before scRNA-seq and barcode sequencing on the cells. **D** UMAP plots showing the $\log_{10}$ normalized gene expression of the drug-susceptible cell marker *SOX10*, and the primed cell markers *EGFR*, *AXL*, and *NT5E*. **E** Diagram of the experimental design and images showing the fixed NT5E-low and NT5E-high cells stained with DAPI after 3 weeks in targeted therapy. Each black dot is the nucleus of a drug-resistant cell, and drug-resistant colonies are circled in black, $n = 3$ biological replicates. **F** UMAP plot showing drug-susceptible cells (in gray) and primed cells (in green).

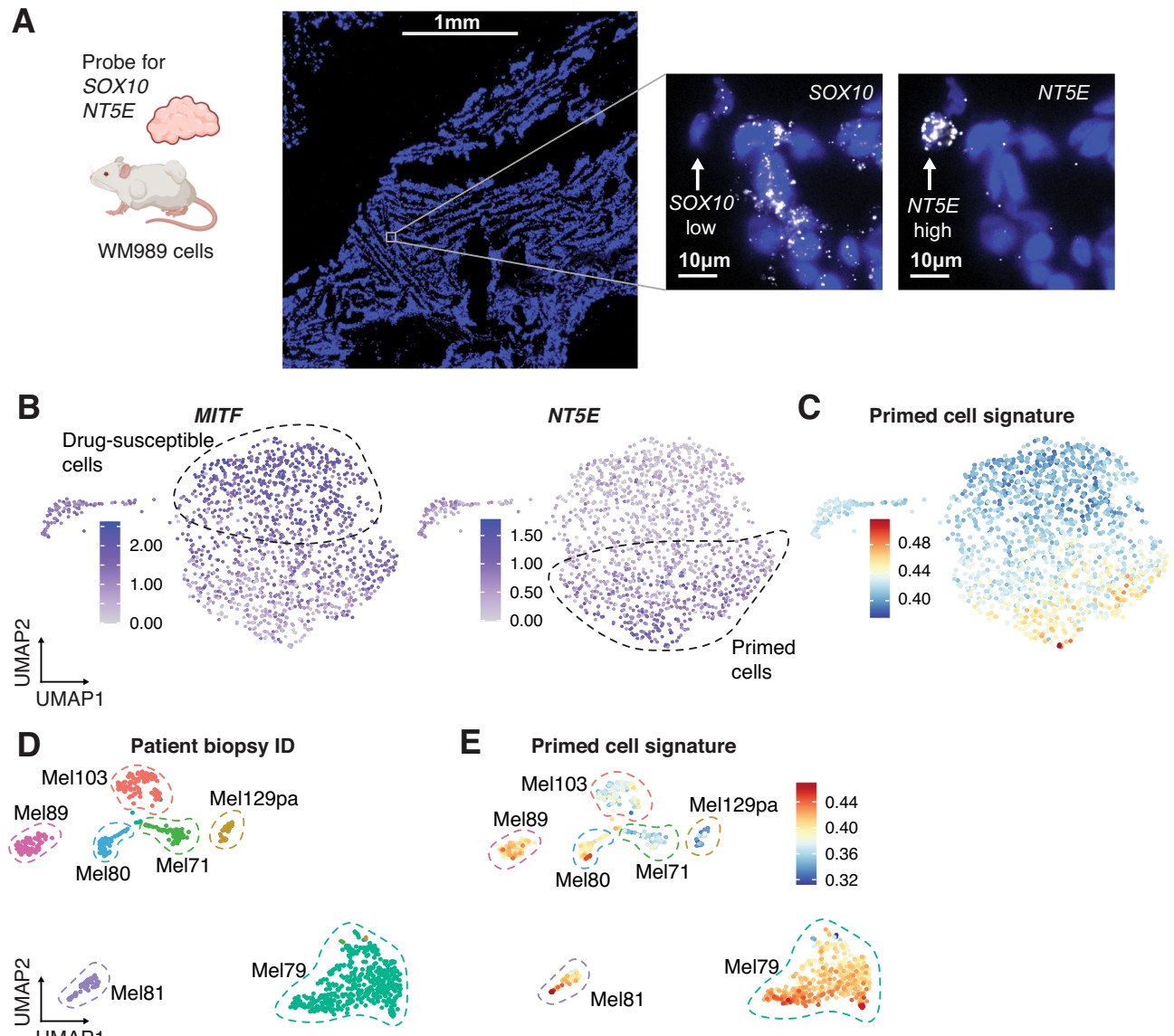

**Fig. 2 | Mouse models and patient samples contain cells expressing primed cell signatures. A** HCR RNA FISH of *NT5E* and *SOX10* in a mouse PDX drug-naive tumor. Nuclei are shown in blue, and RNA for each respective probe is in white. The white arrow points to an example *SOX10*-low *NT5E*-high cell. The leftmost image is a large scan of the tissue section and the scale bar corresponds to 1 mm. The two images to the right are zoomed in on an example cell where the scale bar corresponds to 10 μm, $n = 3$ slides from one PDX tumor. Cartoon created with Biorender.com. **B** scRNA-seq on WM983B melanoma cell line. UMAP plots show the $\log_{10}$

normalized expression of *MITF* and *NT5E* in each cell. **C** UMAP showing WM983B scRNA-seq with each cell colored by the primed gene set signature derived from WM989 cells. **D** Analysis of scRNA-seq of melanoma patient biopsies from[26,27]. The UMAP shows the different biopsies taken from each patient in different colors. **E** UMAP of patient scRNA-seq from (**D**) showing the primed cell gene set signature score. Within each biopsy, some cells have variable expression of genes associated with the primed state.

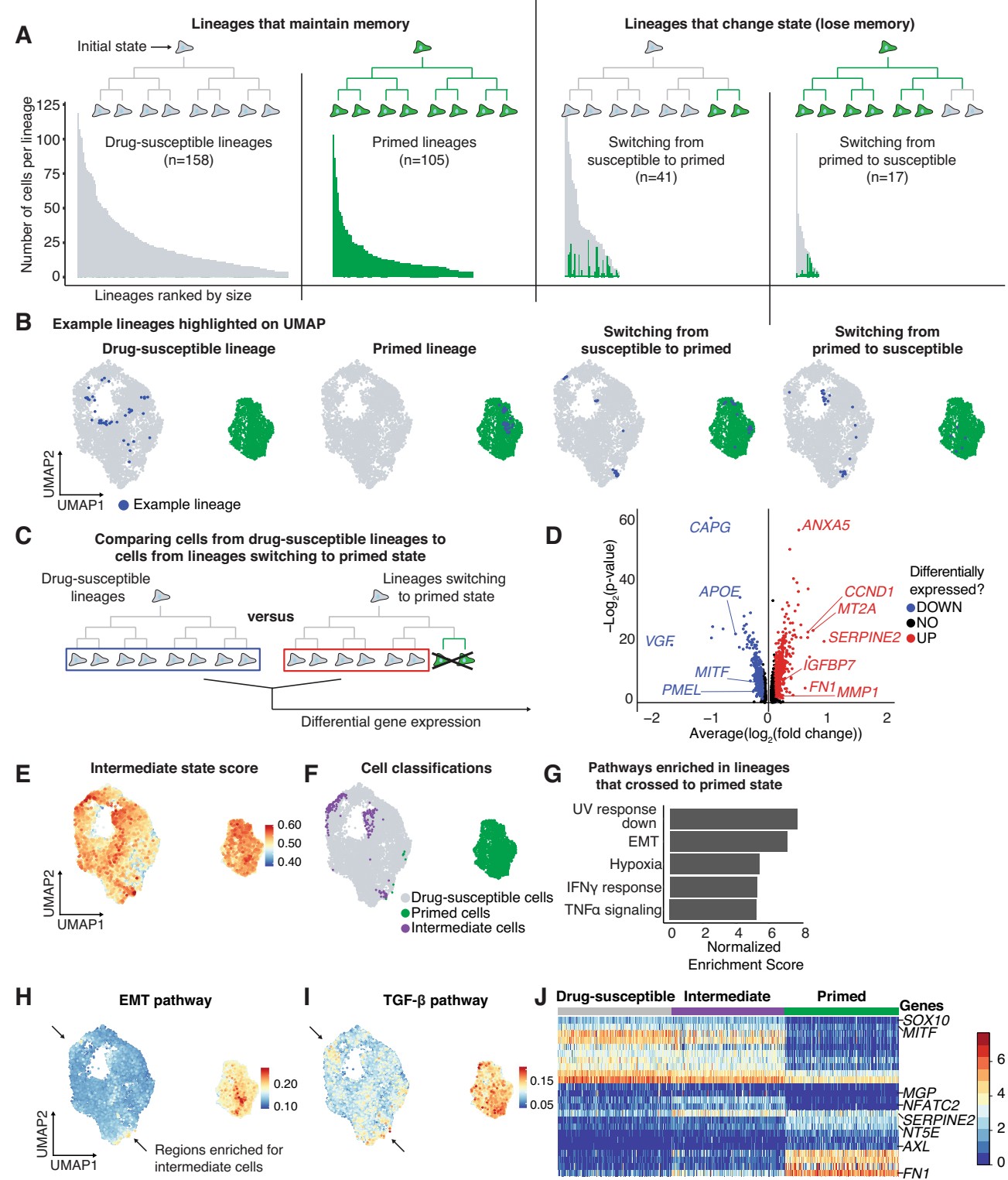

To determine how cell states change over time, we turned to our barcoding data to track lineages. Using the drug-susceptible and primed state clusters from UMAP, we classified each lineage into one of four categories (Fig. 3A, B): (1) Drug-susceptible lineage (containing only drug-susceptible cells), (2) Primed lineage (containing only primed cells), (3) Switching drug-susceptible to the primed state (initial state is drug-susceptible and lineage contains drug-susceptible and primed cells), or (4) Switching from primed to drug-susceptible state (initial state is primed and lineage contains drug-susceptible and

primed cells). While we observe many lineages that contain cells from both states (3 and 4), it is possible that a subset of cells have a higher propensity for state switching and generate these mixed state lineages. Across the classes of lineages, we found that lineage sizes were largest for drug-susceptible lineages and smallest for primed lineages, which is consistent with scRNA-seq predictions that primed state cells divide slower than drug-susceptible cells (Fig. 3A, Supplementary Fig. 3C). Given the high memory observed in both the drug-susceptible and primed states, we compared the differentially expressed genes

**Fig. 3 | An EMT-like state is activated early in the transition to the primed cell state. A** Bar graphs showing the size of each lineage organized based upon classification as drug-susceptible, primed, switching from drug-susceptible to primed, or switching from primed to drug-susceptible. Each bar represents an individual lineage, and the color of the bar indicates the state of the cell (green is primed and gray is drug-susceptible). In lineages that change state, the number of cells in each state is reflected by the colors in the bar. **B** UMAP plots showing an example lineage from each type of lineage in the data. The cells from the example lineage are highlighted in blue. **C** Schematic showing which cells from the drug-susceptible lineages (blue rectangle) and which cells from lineages switching from the drug-susceptible state to the primed state (red rectangle) were used to identify differentially expressed genes in transitioning cells. **D** Volcano plot representing the differential expression analysis outlined in (**C**). Red points represent genes upregulated in crossing lineages and blue points represent genes downregulated in crossing lineages. Wilcoxon rank sum test. **E** UMAP plot showing the gene set signature score of the upregulated genes in panel D (high score represents high expression of the gene set). **F** UMAP plot labeling the top 2% drug-susceptible cells expressing the crossing lineage gene set. We classify these cells as Intermediate state cells, and these cells are labeled in purple. **G** Bar graphs of the normalized enrichment score of the top 5 gene sets enriched in the crossing lineage gene set. **H** UMAP plot showing which cells have a high EMT pathway gene set signature score. Arrows point to drug-susceptible cells with high EMT scores. **I** UMAP plot showing which cells have high TGF-β signaling pathway gene set signature score. Arrows point to drug-susceptible cells with high TGF-β signaling scores. **J** Heatmap of the $\log_{10}$ normalized and scaled gene expression of cells in the drug-susceptible, intermediate, and primed state. The genes shown are the top differentially expressed genes between the primed and drug-susceptible states as well as select genes upregulated in the intermediate state.

between these states captured by scMemorySeq to bulk measurements of gene expression memory performed in the same cell line from Shaffer et al.[2]. We observed a strong correlation between the two methods on a per-gene basis (Supplementary Fig. 3D).

To derive the rates of proliferation and state switching from the paired scRNAseq and barcoding data, we used a stochastic two-state model (Supplementary Methods 1). Using the model, we estimate that primed cells proliferate at approximately half the rate of drug-susceptible cells on average. We also estimate that drug-susceptible cells switch to the primed state once every 135–233 cell divisions, while primed cells are estimated to switch to the drug-susceptible state once every 5–8 cell divisions. The large difference in switching rates is due to the rarity of the primed cell state, which must have a faster switching rate to maintain a constant proportion of primed cells at steady state. Thus, at the single-cell level, the drug-susceptible state is significantly more stable than the primed state.

After classifying the lineages based on whether they switch between states, we next wondered if there are transcriptional differences between lineages that undergo state switching and those that do not. We hypothesized that these differences might exist if there is an intermediate transcriptional state when cells switch between drug-susceptible and primed states. To uncover such a state, we compared the drug-susceptible cells from lineages that contain only drug-susceptible cells to those in lineages that switch from drug-susceptible to the primed state (Fig. 3C). Importantly, to uncover the intermediate state from this comparison requires the assumption that the intermediate state would demonstrate memory through cell division (Supplementary Fig. 3E). Consistent with the presence of such an intermediate state, we found 575 genes differentially expressed between these types of lineages (based upon a differential gene expression analysis with a cutoff of 0.05 Bonferroni adjusted p-value and 0.25 log fold change cutoff) (Fig. 3D, Supplementary Data 1). We then used this gene list to develop an intermediate state score, applied this score to all cells in the data set, and then projected them into UMAP space (Fig. 3E). We identified two primary regions within the drug-susceptible cluster that are enriched for cells in this intermediate expression state. We set a threshold on this score and classified the high-scoring cells as intermediate state cells (Fig. 3F). These rare intermediate state cells are confidently identified using lineage information, but are difficult to accurately identify from the scRNAseq alone (Supplementary Fig. 1G).

We next wanted to know what pathways are activated in the intermediate state. We first used gene set enrichment analysis on the gene list from the intermediate state (Fig. 3D). The top pathways included UV response down, epithelial-to-mesenchymal transition (EMT), and response to hypoxia (Fig. 3G). We examined the activity of these different pathways in UMAP space and noted that the EMT pathway score localized in the same regions of the UMAP as the intermediate state cells (Fig. 3H). We focused on EMT as multiple lines of evidence in melanoma suggest that an EMT-like gene expression

state is associated with resistance to BRAFi/MEKi[16,30–33]. Another enriched pathway in our analysis was TGF-β signaling, which also showed enrichment in the same region of the UMAP as the intermediate state cells (Fig. 3I) and is associated with resistance in the literature[20,32]. Furthermore, both the EMT and TGF-β pathways were enriched in the primed state beyond the levels in intermediate state cells, suggesting that these are early changes as cells switch from drug-susceptible to the primed state (Fig. 3H, I). We also observed that these intermediate state cells maintain expression of many genes associated with the drug-susceptible state, including melanocyte identity genes *SOX10*, and *MITF*, but have already begun to upregulate some of the important primed state marker genes including *FN1* and *SERPINE2* (Fig. 3J). A few genes were uniquely expressed only in the intermediate state cells, including *NFATC2*, and *MGP*. Intriguingly, *NFATC2* was previously found to be a regulator of *MITF* and melanoma dedifferentiation[34,35]. Together, these data suggest that the intermediate state represents the initiation of an EMT-like process as cells switch states.

Given the role of TGF-β as a potent inducer of EMT and the enrichment of the TGF-β pathway in the intermediate state, we hypothesized that applying TGFB1 to drug-susceptible cells would induce the primed state (Fig. 4A)[36]. In addition, we observed that TGFB1 and its receptor are highly upregulated in primed state cells, suggesting that these cells can activate TGF-β signaling through autocrine or paracrine mechanisms (Supplementary Fig. 3F). To test the hypothesis that TGFB1 induces the primed state, we treated melanoma cell lines, including WM989 and WM983B, with recombinant TGFB1 for 5 days and then performed flow cytometry for primed cell marker gene NT5E (Fig. 4B). We found that treatment with TGFB1 increased the percentage of cells in the primed state in WM989 from 1.98 to 19.15% and in WM983B from 10.04% to 81.56% (Fig. 4C, D, Supplementary Fig. 4C, D). These findings are in agreement with the literature showing that TGF-β signaling can induce a dedifferentiation state in melanoma[20,37]. The dedifferentiated state is similar to our primed state, as these cells are also characterized by their decreased expression of melanocyte transcription factors *SOX10* and *MITF*[20,26,37,38].

Since TGFB1 was sufficient to induce the primed state, we next asked whether inhibiting TGF-β signaling could induce the drug-susceptible state. We treated WM989 cells with LY2109761, a targeted TGFBR1/2 inhibitor (TGFBRi), for 5 days. Unexpectedly, we found that the TGFBRi did not have a large impact on the percentage of cells in the primed state (Fig. 4C). To confirm the specificity of the inhibitor, we treated WM989 cells with both recombinant TGFB1 and TGFBRi and found that indeed the inhibitor was able to block the effects of TGFB1 (Fig. 4C). Overall, this suggests that TGF-β signaling is sufficient to increase the percentage of primed state cells, but is not necessary for maintaining the primed expression state.

Since inhibiting TGF-β signaling was not able to switch cells out of the primed state, we wondered whether other signaling pathways might be involved in maintaining the primed state. We noted that several growth factors including *FGF1, VGF, BDNF, VEGFA, VEGFC,* and

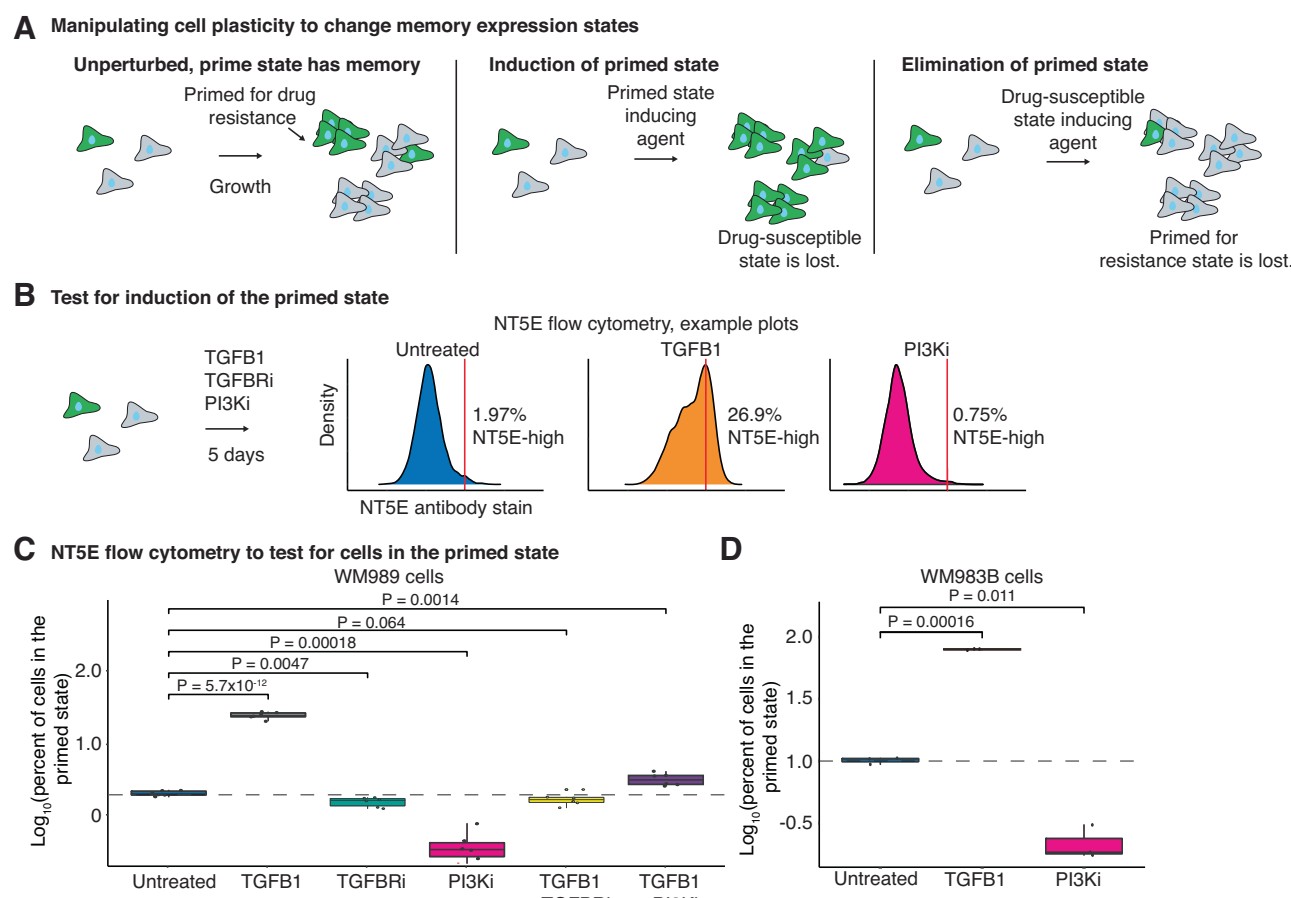

**Fig. 4 | TGFB1 and PI3K inhibition can modulate the number of primed cells in the population. A** Schematic showing the expected number of primed cells over time if cells are untreated, treated with a drug or protein that induces the primed state, or treated with a drug or protein that disrupts the primed state. **B** Example flow cytometry density plots from NT5E stained WM989 cells after 5 days of their respective treatments. **C** Box plot of the $\log_{10}$ percent of WM989 cells in the primed state after 5 days in each treatment. All conditions have four biological replicates each with six technical replicates. $P$ values were calculated using a two-sided $t$-test. The dotted line represents the mean $\log_{10}$ percent of primed cells (set at 2%) in the untreated condition. **D** Box plot of the $\log_{10}$ percent of WM983B cells in the primed state after 5 days of each treatment. All conditions have two biological replicates each with three technical replicates. $P$ values were calculated using a two-sided $t$-test. The dotted line marks the mean $\log_{10}$ percent of primed cells (set at 10%) in the untreated condition. For the box plots in (**C**) and (**D**) the center line is the median, the box is the interquartile range (IQR), and the whiskers indicate 1.5 times the upper and lower IQR. All data points are shown.

*PDGFA* were all upregulated in the primed state (Supplementary Data 2). Many of these growth factors have downstream PI3K activation as a common pathway[39]. We also noted that while the canonical TGF-β signaling has downstream SMAD family effectors[40], non-canonical TGF-β signaling can activate PI3K[41,42]. Since these pathways converge on PI3K, we wondered whether blocking PI3K could eliminate the primed state by blocking both the effects of TGF-β signaling and other upregulated growth factors and receptors. To test this hypothesis, we treated WM989 and WM983B cells with a PI3K inhibitor (PI3Ki), GDC-0941, for 5 days and then performed flow cytometry looking at NT5E to quantify the percentage of cells in the primed state. Importantly, we selected the dose of 2μM for the PI3Ki as it has minimal effects on cell viability and growth rate and effectively blocks PI3K signaling (Supplementary Fig. 4A, B, E). We found that the PI3Ki decreased the percentage of cells in the primed state in WM989 and WM983B from 1.98 to 0.31% and 10.04 to 3.66% respectively (Fig. 4C, D). Furthermore, we tested whether the PI3Ki can block the effects of TGF-β signaling by simultaneously treating WM989 cells with TGFB1 and PI3Ki. Indeed, we found that the PI3Ki was able to block the increase in primed state cells seen when we treat with TGFB1 alone (Fig. 4C), suggesting that the TGFB1-mediated effects on priming melanoma cells is dependent on downstream PI3K activity.

Because PI3Ki was able to reduce the percentage of primed state cells, we wonder whether other ligands that can activate the PI3K pathway could induce the primed state[43–45]. We treated WM989 cells with EGF, BDNF, and IL6 for 5 days (Supplementary Fig. 4F) and found that none of these factors were able to induce the primed cell state as seen with TGFB1. Thus, we concluded that TGFB1 is a unique inducer of the primed state, potentially due to its ability to stimulate the PI3K and SMAD signaling pathway at the same time[46,47].

Since TGFB1 and the PI3Ki were both able to change the percentage of cells in the primed state, we wondered how this is achieved at the single-cell level. Specifically, does TGFB1 force more cells to switch into the primed state? Conversely, does the PI3Ki force cells to exit the primed state? Based on our flow cytometry experiments, we can conclude that the percentage of the population is shifted by these treatments; however, changes in multiple different parameters could lead to this same effect. For instance, the result that TGFB1 increases the percentage of primed cells could be explained by (1) an increase in the growth rate of primed cells, (2) a selective killing of drug-susceptible cells, or (3) state switching from the drug-susceptible into the primed state. The opposite consideration is necessary with the finding that the PI3Ki decreases the percentage of primed cells. This result could be the effects of (1) an increase in growth rate in the drug-

susceptible cells, (2) selective killing of primed cells, or (3) state switching from the primed into the drug-susceptible state.

To directly test how TGFB1 and PI3K shift the percentage of primed state cells, we used the fact that these cell states have memory to test the effects of these perturbations on cells in each state. Specifically, we transduced WM989 cells with the high-complexity transcribed barcode library and then allowed the cells to go through 7–8 doublings (as we expect the memory of these states to largely persist on this timescale, Supplementary Fig. 5A–C). We then split the cells into four separate plates and subjected each to a different condition (untreated, TGFB1, TGFBRi, and PI3Ki) (Fig. 5A). Since cells within the same lineages have very similar gene expression states due to memory, this allows us to approximate studying how the same cell will react to the different conditions. After 5 days, we harvested each sample and performed scRNA-seq and barcode-sequencing to capture the transcriptional state and barcode under each condition (Fig. 5A). With this experimental design, each barcode is represented across all the conditions. Thus, we can see the effects of each treatment by comparing the gene expression of cells with a given barcode in one treatment to cells with the same barcode in the untreated condition.

We analyzed a total of 40,021 cells across the four different samples and found similar transcriptional states distributed across UMAP space, as in previous experiments (Fig. 5B, Supplementary Fig. 6A). To classify cells as primed or drug-susceptible, we selected Louvain clusters 4 and 10, as these clusters contained the majority of cells expressing the known primed state marker genes (Fig. 5D, Supplementary Fig. 6B, C). Based on this classification, we see that TGFB1 increased the percent of cells in the primed state and that PI3Ki decreased the percent of cells in the primed state (Fig. 5C, E). Importantly, the primed cell state induced by treatment with TGFB1 was transcriptionally very similar to untreated cells in the primed state, and induced EMT and TGF-β signaling genes (Fig. 5F, Supplementary Fig. 6D, E). To explicitly test for differences, we performed differential gene expression to compare the TGFB1-induced and untreated primed state cells. We found that some genes, including *NGFR*, *FGFR1*, *FOSL1*, and *JUN*, were induced by the TGFB1 treatment, but were not as highly expressed as in the untreated primed state cells (Fig. 5F, Supplementary Fig. 6F). This might be significant as *NGFR* expression has been linked to invasive properties of melanoma cells[48,49]. We also noted that TGFB1 seemed to induce even higher expression for many of the primed state genes including *FN1*, *SERPINE1*, *COL1A1*, and *VGF* (Fig. 5F, Supplementary Fig. 6F). Some of these effects might result from the specific dose of TGFB1, which is unlikely to be the same as the amount of TGFB1 found endogenously. Overall, this analysis shows that TGFB1 and PI3Ki change the whole transcriptome of cells into the primed and drug-susceptible state respectively.

Next, we integrated the barcoding data into the analysis to test for state switching at the single-cell level. Our data set included 19,740 lineages, each containing a minimum of 3 cells per lineage, with 49% of lineages represented across all four conditions. To test the hypothesis that TGFB1 causes cells to switch to the primed state, we first evaluated lineages where the untreated sample consisted solely of cells in the drug-susceptible state. Across each of these lineages, we quantified the fraction of primed cells in the other conditions and found that the matched set of lineages in the TGFB1-treated condition had a higher fraction of cells in the primed state (Fig. 5G, Supplementary Fig. 6G, H). Reassuringly, the TGFBRi and the PI3Ki only had minor increases in the percentage of primed state cells across these lineages. We next tested the opposite direction, switching from the primed state to the drug-susceptible state. We identified lineages in which the entire lineage was in the primed state in the untreated sample, 8 total. In the PI3Ki-treated sample, 6 out of these 8 lineages had cells that had switched out of the primed state (Fig. 5H, Supplementary Fig. 6G, H). Of note, 2 lineages did not respond to the PI3K. However, by analyzing all lineages, we find that 93% of lineages reduce their fraction of primed state cells when

treated with PI3Ki (Supplementary Fig. 6G). Lineages that did not respond may require longer treatment with the PI3Ki to switch to the drug-sensitive gene expression state. Moreover, there were no systematic gene expression differences that explained the differences in responsiveness to PI3Ki (Supplementary Data 3). Taken together, these data show that TGFB1 and the PI3Ki can induce state switching and that the observed changes in the number of primed state cells are not due to other population dynamics.

To extend our analysis to include all of the lineage data, we developed a stochastic model of state switching between the drug-susceptible and primed states. Our model included different state-switching parameters ($k_{on}$, $k_{off}$), different growth rates, and different death rates in each cell state (Supplementary Fig. 6I). We used the model to simulate experimental data for different scenarios in which different parameters are changing. For instance, for TGFB1, the increase in primed state cells could be explained by three possible parameter changes, (1) an increase in $k_{on}$ for the primed state, (2) an increase in primed cell proliferation rate, or (3) an increase in the death rate among drug-susceptible cells. To constrain the proliferation rate parameter in our model, we performed live-cell imaging to measure the direct effects of 5 days of TGFB1 and the PI3Ki on the proliferation rates of primed and drug-susceptible cells. Across conditions, we found that cells in the primed state proliferate more slowly compared to the drug-susceptible cells (Supplementary Fig. 6J). In addition, we found that treatment with either TGFB1 or the PI3Ki decreased growth rate by similar amounts in each population (Supplementary Fig. 6J).

Given these constraints on proliferation rates, we then ran one million simulations of the model varying each parameter and found that increasing the rate at which the cells switch into the primed state is the best fit for our data. In addition, this model suggests that drug-susceptible cells in lineages with a high proportion of cells already in the primed state are more easily switched into the primed state (See Supplementary Methods 1). We similarly considered possible parameter changes that could account for the decrease in primed state cells upon PI3Ki treatment. Given experimentally measured constraints on proliferation rates, we found that an increased rate of cells switching from the primed state to the drug-susceptible state was the best fit for our data (See Supplementary Methods 1). This model further validates our finding that the effect of TGFB1 and PI3Ki on the number of primed state cells in the population works through state switching.

Given that TGFB1 and PI3Ki are drivers of state switching, we hypothesized that these treatments would also impact drug resistance. We used a sequential dosing strategy in which we first pretreat cells for 5 days with a state modulator (e.g., TGFB1 or PI3Ki) followed by 4 weeks of BRAFi/MEKi to test for subsequent resistance (Fig. 6A). In this design, the pretreatment period is intended to induce state switching before the addition of targeted therapy. When we pretreated cells with TGFB1, we found that, although fewer total cells died, pre-treatment with TGFB1 initially led to faster killing of the remaining drug-susceptible cells compared to cells treated with BRAFi/MEKi alone (Fig. 6B, C, Supplementary Fig. 7A). Furthermore, after 4 weeks of BRAFi/MEKi, we found that the pretreated TGFB1 sample had fewer resistant cells than the BRAFi/MEKi only sample. Given the potential for the pretreatment time window to affect cell growth, we normalized the resistant data by the number of cells present after pretreatment in each condition. With this normalization, we found that TGFB1 increased resistance by 2.8-fold compared to the control (Supplementary Fig. 7B, C). We also tested pretreating with TGFBRi and, consistent with its minimal effects on priming, we found that it did not change the amount of resistance (Fig. 6B–D).

We then tested the more therapeutically relevant approach of pretreating with the PI3Ki to reduce the number of primed cells, followed by treating with BRAFi/MEKi. We found that pretreating with PI3Ki decreased the number of resistant colonies by 62% and the

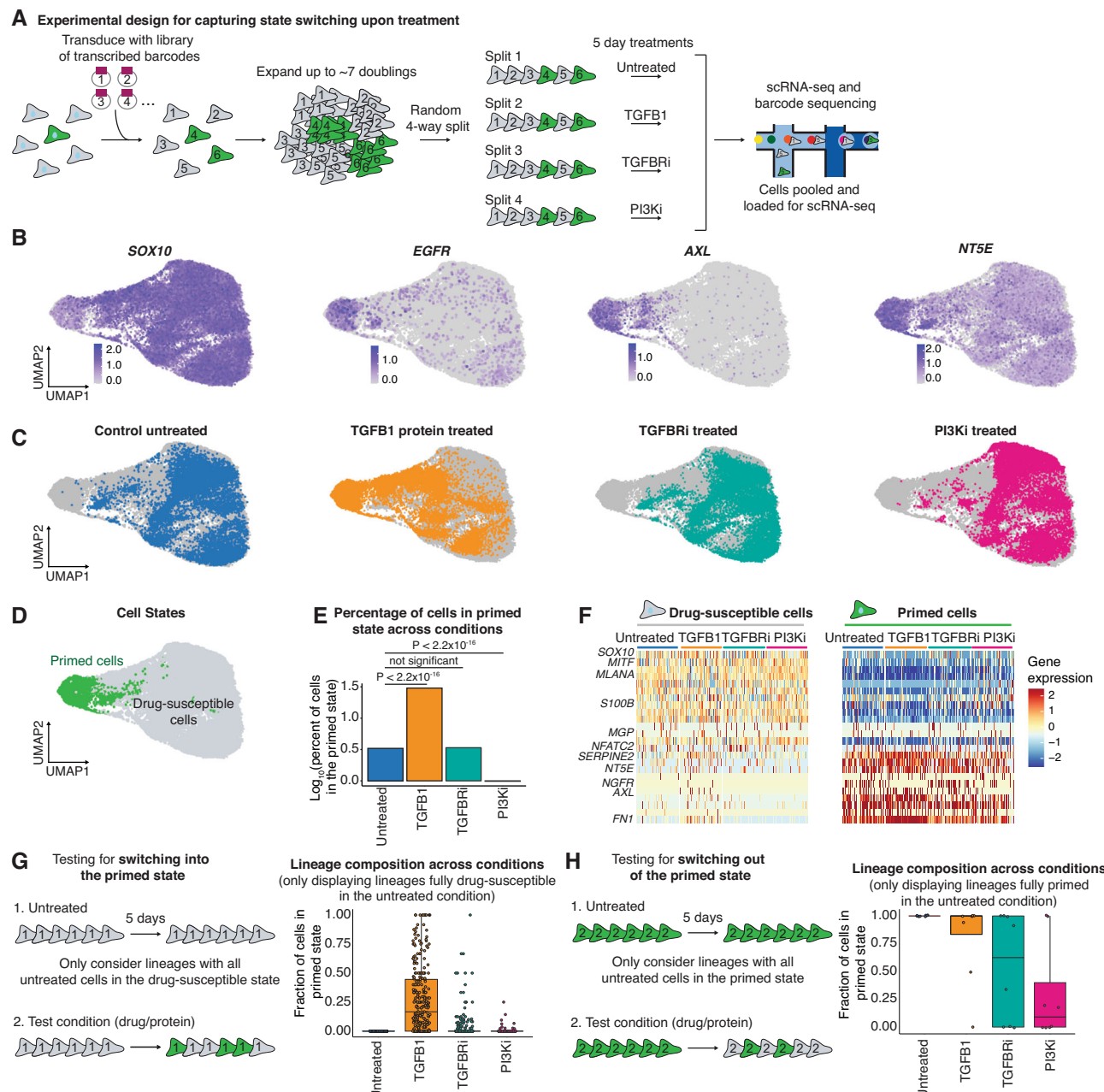

**Fig. 5 | Treatment with TGFB1 induces the primed state and treatment with PI3K inhibitor induces the drug-susceptible state. A** Schematic of the experimental design used to determine if treatments cause state switching. We transduced melanoma cells with barcodes, allowed them to divide ~7 times, and split them across four plates. Cells with the same barcodes serve as copies of each other. We used scRNA-seq with barcode sequencing to capture both the lineage and transcriptome of the cells at the endpoint. **B** UMAP plots of the $\log_{10}$ normalized gene expression of the drug-susceptible state marker *SOX10* and primed state markers *EGFR*, *AXL*, and *NT5E*. **C** UMAP plots highlighting cells in each condition relative to all the other sequenced cells (in gray). Cells in blue were untreated, cells in orange were treated with TGFB1, cells in teal were treated with TGFBRi, and cells in pink were treated with PI3Ki. **D** UMAP plot with primed cells labeled in green and drug-susceptible cells in gray. **E** Bar graph quantifying the $\log_{10}$ percent of primed cells in each condition based on the defined drug-susceptible and primed cell

populations in (**D**) and the location of each treatment condition shown in (**C**). *P* values were calculated using Pearon's Chi-squared test with Yates' continuity correction. **F** Heatmaps of $\log_{10}$ normalized and scaled gene expression of drug-susceptible and primed cells in each treatment. **G** Schematic of lineage-based analysis to test for state switching into the primed state. Box plots show the fraction of cells in each lineage that are in the primed state. The lineages shown are exclusively those that were completely drug-susceptible in the untreated sample. **H** Schematic of lineage-based analysis to test for state switching into the drug-susceptible state. Box plots show the fraction of cells in each lineage that are in the primed state. The lineages shown are exclusively those that were completely primed in the untreated sample. For the box plots in (**G**) and (**H**), the center line is the median, the box is the IQR, and the whiskers indicate 1.5 times the upper and lower IQR. All data points are shown.

number of resistant cells by 57% compared to BRAFi/MEKi alone (Fig. 6C, D). We next performed the same normalization as described above for TGFB1. We found that with this normalization, PI3Ki pretreatment decreased the number of resistant cells by 36% (Supplementary Fig. 7B, C). Furthermore, treating cells with PI3Ki and targeted

therapy at the same time was even more effective than pretreatment, nearly eliminating all resistance (Supplementary Fig. 7D, E). Although blocking the PI3K pathway at the same time as the MAPK pathway is effective at killing melanoma cells, it is toxic to patients which has forced clinical trials to use low doses of these drugs, thus limiting the

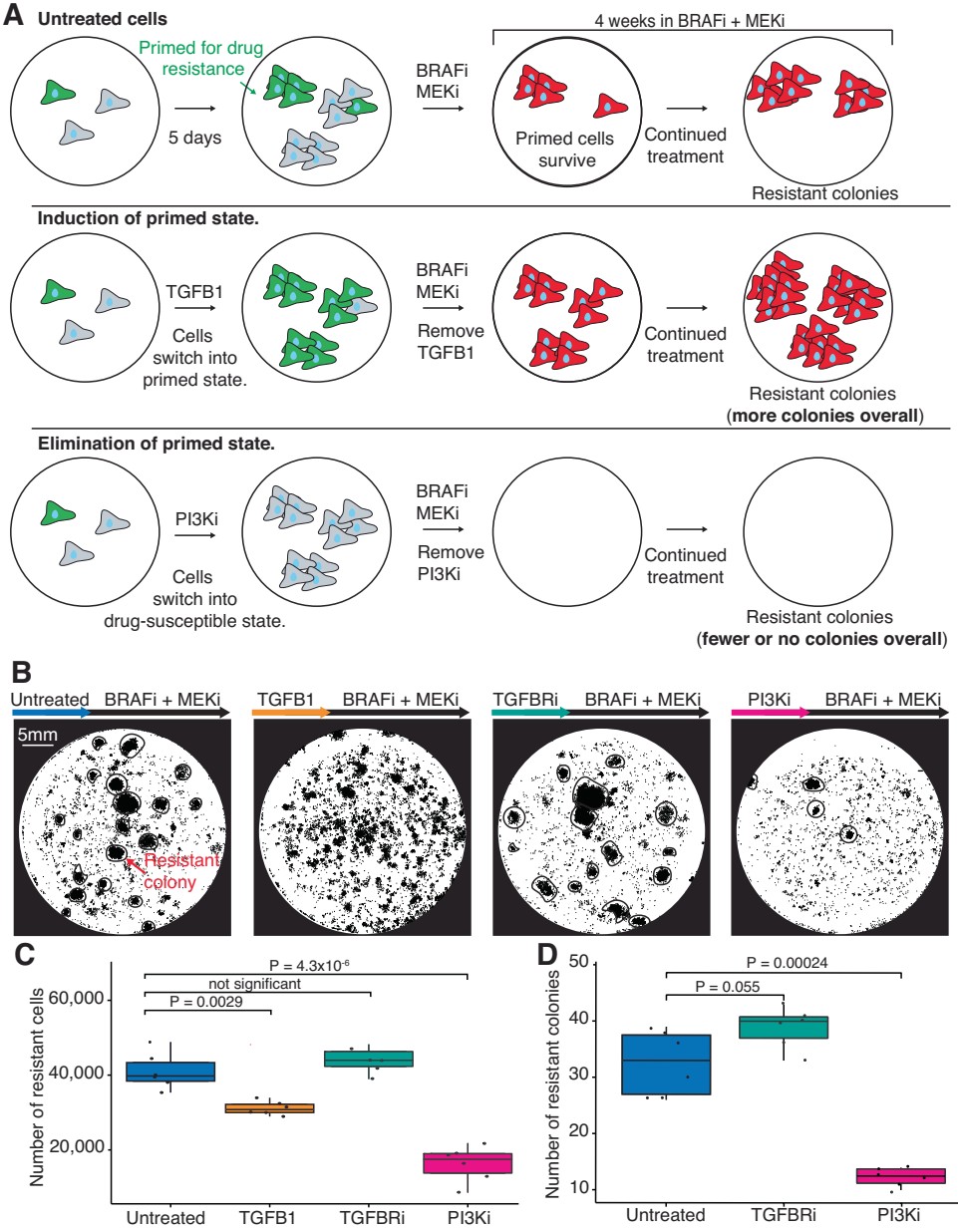

**Fig. 6 | Decreasing the number of primed cells results in a decrease in resistance. A** Schematic of the experimental design testing whether modulators of state switching affect the number of drug-resistant colonies (represented by red cells). In this experiment, we treated cells with the state-switching treatments for 5 days, removed the treatment, and then treated them with BRAFi/MEKi for 4 weeks. After 4 weeks we quantified drug resistance by counting the number of drug-resistant cells and the number of drug-resistant colonies. **B** Example images of fixed cells stained with DAPI for each condition after 5 days of pretreatment and 4 weeks of treatment with BRAFi/MEKi. Drug-resistant colonies are circled in black where readily identifiable. The scale bar in the first image represents 5 mm and applies to all images in the panel. **C** Box plot quantifying the number of drug-resistant cells from scans like those shown in panel (**B**) with six technical replicates. *P* values calculated using a two-sided *t*-test, *n* = 3 biological replicates. **D** Box plot quantifying the number of drug-resistant colonies (for each condition in which distinct colonies were visible) from images like those shown in panel (**B**) with six technical replicates. *P* values were calculated using a two-sided *t*-test, *n* = 3 biological replicates. For the box plots in (**C**) and (**D**) the center line is the median, the box is the IQR, and the whiskers indicate 1.5 times the upper and lower IQR. All data points are shown.

effectiveness of this otherwise promising combination[50–52]. In sum, we find that by first converting primed cells to drug-susceptible cells through pretreatment with PI3Ki, we can reduce resistance to targeted therapy and potentially provide some of the benefits of combination therapy, while minimizing toxicity.

## Discussion

Here, we show that scMemorySeq is a powerful method for tracking gene expression memory in single cells. Our approach leverages the combination of scRNA-seq and cellular lineage barcoding to quantify memory of gene expression states in single-cell data. We applied this method to melanoma cells to track lineages as they switch states between a drug-susceptible state and a state primed for drug resistance. By analyzing the gene expression differences in lineages that switch states, we identified and tested TGF-β and PI3K as mediators of state switching at the single-cell level. Ultimately, we show that by manipulating state switching, we can reduce resistance to targeted therapy.

Broadly, it is intriguing that modulating signaling alone is sufficient to globally modify gene expression states in single cells and to

affect their susceptibility to drugs. Here, PI3Ki is driving cells into a MAPK-dependent transcriptional state, sensitizing cells to MAPK inhibitors. Importantly, multiple papers have reported the use of PI3Ki to reduce resistance in melanoma[53–56] and other studies have proposed leveraging cell state dynamics in therapy scheduling[57–59]. Distinguishing our use of a PI3Ki, here, we find that the PI3Ki can reduce drug resistance even when only applied briefly before the addition of targeted therapy. This approach may lay out a generalizable strategy for reducing drug resistance in which perturbing signaling can globally tune gene expression to achieve susceptibility to drugs. In such a strategy, modulating signaling pathways would be used before the addition of the main targeted therapy to drive heterogeneous populations of cells into a drug-susceptible state. This is in contrast to dosing with a combination of inhibitors at the same time, which is not always tolerated due to toxicity and side effects[60–62].

Conceptually, the idea of leveraging the plasticity of cells to revert them into a drug-susceptible state to delay drug resistance has previously been described in the context of drug holidays[63]. In contrast to our approach, in which we actively drive cells into a drug-susceptible state, a drug holiday is a break from the drug to alleviate the selective pressure thus allowing the drug-resistant cells to switch back to a drug-susceptible state at their intrinsic rate. Given the previous success of drug holidays[63,64], one strategy that we believe should be further explored is to use state-switching drugs during what would be the holiday period. With this approach, the switch to a drug-susceptible state would be accelerated during the drug holiday, leading to potentially even less resistance. Future work is still needed to model these different scenarios and to experimentally test the efficacy of such dosing strategies.

Extending scMemorySeq beyond state switching in melanoma, there are several biological contexts in which specific drugs or ligands could be shifting gene expression states and population dynamics simultaneously. This is particularly relevant in cancer where there is a growing body of literature describing considerable heterogeneity at the single-cell level that shows variable degrees of memory[26,65,66]. Here, we show that the scMemorySeq approach can provide a detailed systematic overview of these populations, and can also be used to test how closely related cells from the same lineage will react to different conditions. We believe that this approach is generalizable and could be used in other contexts to profile cell state transitions under different drugs, ligands, or environmental conditions. This is a growing area of interest as multiple recent studies showed that microenvironment and growth conditions of cancer cells can globally change both gene expression and sensitivity to drugs[66–68].

In sum, we show how scMemorySeq uses cellular barcoding to reveal single-cell dynamics of drug resistance in melanoma. By tracking the memory of gene expression states we can identify stable cell populations as well as the factors that cause cells to change states. This approach can be widely applied to discover unknown dynamics in heterogeneous cell populations and to identify the key factors responsible for gene expression state changes in biological systems.

## Methods

### Ethics statement
This study complies with all relevant ethical and safety regulations. The PDX tissue was collected by the Weeraratna lab as part of a prior study[6], further described in the section titled Mouse model tumor generation. All animal experiments were approved by the Institutional Animal Care and Use Committee (no. 112503X_0) and were performed in a facility accredited by the Association for the Assessment and Accreditation of Laboratory Animal Care.

### Antibodies
Antibodies used in this included NGFR primary antibody (1:11, Biolegend, 345108), EGFR primary antibody (1:200 dilution, Fisher

Scientific, Clone 225, MABF120MI), NT5E primary antibody (1:200, Biolegend, 344005), Alexa Fluor 488 Donkey Anti-Mouse secondary antibody (1:500, Jackson Labs, 715-545-151), Phospho-AKT Ser473 (1:1000, Cell Signaling Technology #4060), pan-AKT (1:1000, Cell Signaling Technology #9272), β-actin (1:1000, Santa Cruz Biotechnology #4778).

### Small molecule inhibitors and recombinant proteins
Vemurafenib (Selleckchem, S1267) was reconstituted at 4 mM in DMSO for the stock solution. Dabrafenib (Cayman, 16989-10) was reconstituted at 1.25mM in DMSO for the stock solution. Trametinib (Cayman, 16292-50) was reconstituted at 12.5 μM in DMSO for the stock solution. TGFB1 (R&D systems, 240-B-002) was reconstituted at 100 μg/mL in a 4mM hydrochloric acid, 1 mg/mL bovine serum albumin solution for the stock. PI3K inhibitor (GDC-0941, Cayman, 11600-10) was reconstituted at 10 mM in DMSO for the stock solution. TGFBRi (LY2109761, SML2051-5MG) was reconstituted at 40 mM in DMSO for the stock solution. IL6 (R&D systems, 206-IL-010) was reconstituted at 100 μg/mL in a 0.1% BSA PBS solution for the stock. EGF (R&D systems, 236-EG-200) was reconstituted at 200 μg/mL in PBS for the stock solution. BDNF (R&D systems, 206-IL-010) was reconstituted at 100 μg/mL in a 0.1% BSA PBS solution for the stock. EGF (R&D systems, 248-BDB-005) was reconstituted at 100 μg/mL in water for the stock solution.

### Cell lines and tissue culture
We used the following cell lines: WM989 A6-G3, which are a twice single-cell bottlenecked clone of the melanoma line WM989 (provided by the Meenhard Herlyn's lab at the Wistar Institute), WM983B E9-D5, which are a twice single-cell bottlenecked clone of the melanoma line WM983B (provided by the Meenhard Herlyn's lab at the Wistar Institute), and HEK293FT cells which we used for lentiviral packaging (provided by Arjun Raj's lab at the University of Pennsylvania). We authenticated the identity of all cell lines by STR profiling and confirmed that they are all negative for mycoplasma. STR profiling and mycoplasma testing were performed by the Penn Genomic Analysis Core. We cultured WM989 A6-G3 and WM983B E9-D5 in TU2% (78.4% MCDB 153, 19.6% Leibovitz's L-15, 2% FBS, 1.68 mM CaCl, 50 Units/mL penicillin, and 50μg/mL streptomycin). We cultured HEK293FT in DMEM 5% (95% DMEM high glucose with GlutaMAX, 5% FBS, 50 Units/mL penicillin, and 50 μg/mL streptomycin). We grew all cells at 37 °C and 5% $CO_2$ and passaged them using 0.05% trypsin-EDTA.

### Barcode library
We used a high-complexity transcribed barcode library described in Emert et al. for our lineage barcodes[10]. The plasmid uses LRG2.1T as a backbone, but we replaced the U6 promoter and sgRNA insert with GFP followed by a 100 nucleotide semi-random barcode, expressed by an EFS promoter. The barcode is semi-random as it is made up of WSN repeats (W = A or T, S = G or C, N = any) to maximize barcode complexity. A detailed protocol on the barcode production process can be found in Emert et al., which also links to this protocol:

https://www.protocols.io/view/barcode-plasmid-library-cloning-4hggt3w.

The sequence of the plasmid can be found here:

https://benchling.com/s/seq-DAMUWPyU198hRSbpiecf?m=slm-GJ609ijArVWmkT8mk8zr.

### Lentiviral packaging
We grew HEK293FT to about 90% confluence in a 10cm dish containing 10 mL of media (see the section titled Cell lines and culture). To transfect cells with the lentiviral plasmid, we combined 500μl of OPTI-MEM with 80 μl of 1 mg/mL PEI in one tube. In a second tube, we combined 500μL of OPTI-MEM, 9 μg of the psPAX2 plasmid, 5.5 μg of the VSVG plasmid, and 8 μg of the barcode plasmid. We combined the contents of these two tubes and allowed the mixture to incubate at

room temperature for 15 min. We then pipetted the solution dropwise into the plate of HEK293FT and incubated the cells at 37 °C for 7 h. Next, we removed the media, washed the plate once with DPBS, added 10 mL of fresh media, and then incubated the cells at 37 °C for ~12 h. We used fluorescence microscopy to confirm GFP expression in the cells and then applied a fresh 6 mL of media for virus collection. We incubated the cells at 37 °C for ~12 h and collected the media (this media contains the virus). We repeated the process of adding 6mL of media and collecting the virus every 12 h for a total of ~72 h. After the last collection, we filtered all the media containing the virus through a 0.2 μm filter to ensure no HEK293FT cells were left with the virus media. Finally, we made 1mL aliquots of the media containing the virus and stored them at −80 °C.

### Lentiviral transduction

When barcoding cells, we wanted to avoid multiple lineage barcodes per cell, and thus, we aimed to transduce ~20% of cells. To transduce the cells, we made a mixture of polybrene (4μg/mL final concentration), virus (concentration determined through titration experiments to achieve 20% infection), and cells at 150,000 cells/mL. Next, we put 2 mL of this mixture into each well of a 6-well plate and spun the plate at 600 RCF for 25 min. We then incubated the cells with the virus at 37 °C for 8 h. After the incubation, we removed the media containing the virus, washed each well with DBPS, and added 2mL of fresh media to each well. The next day, we transferred each well to its own 10 cm dish. We then gave the cells 2–3 days to start expressing the barcodes. We confirmed the presence of barcodes by GFP expression in the cells (cells express GFP along with the barcode).

### Fluorescence-activated cell sorting (FACS)

We dissociated cells into a single-cell suspension using trypsin-EDTA and washed them once with 0.1% BSA. To stain for EGFR and NGFR, we first stained with the EGFR antibody (see antibodies section) diluted 1:200 in 0.1% BSA for 1 h on ice. We then washed the cells twice with 0.1% BSA and stained them with the anti-mouse A488 secondary antibody at a 1:500 dilution in 0.1% BSA for 30 min on ice. Next, to stain for NGFR, we washed the cells once with 0.1% BSA and then resuspended them in a 1:11 dilution of the NGFR antibody directly conjugated to APC in 0.5% BSA 2 mM EDTA solution. We then incubated the cells on ice for 10 min. Finally, we washed the cells once with 0.5% BSA 2 mM EDTA, resuspended them in 1% BSA with DAPI, and kept the cells on ice until sorting.

To stain for NT5E, we resuspend the cells in a solution of NT5E antibody diluted 1:200 in 0.1% BSA and incubate them on ice for 30 min. We then washed the cells twice with 0.1% BSA, resuspended the cells in 1% BSA with DAPI, and kept them on ice until sorting.

For flow sorting, we followed the staining protocols above and then sorted the cells on a Beckman Coulter Moflo Astrios with a 100 μm nozzle using Summit software (Version 62). We used forward and side scatter to separate cells from debris and select singlets. We selected DAPI-negative cells to remove dead cells. To sort primed cells with EGFR and NGFR, we selected the top 0.2% of EGFR and NGFR-expressing cells. To sort primed cells with NT5E, we selected the top 2% of NT5E-expressing cells.

### Single-cell RNA sequencing

We used the 10x Genomics 3′ sequencing kits for all our scRNA-seq experiments. For the first scRNA-seq experiment, introduced in Fig. 1C, we sorted 1000 barcoded WM989 cells per well in a 96-well plate (one well of mixed cells and one of EGFR/NGFR-high cells) and allowed them to expand through 4–5 doublings. We then trypsinized one mixed well and one primed well and prepared as described in the Chromium Single Cell 3′ Reagent Kit V3 user guide. When loading cells on the microfluidic chip, we split both the primed and the mixed cells across 2 wells. After GEM generation, we continued to follow the Chromium Single Cell 3′ Reagent Kit V3 user guide to generate libraries.

For the second scRNA-seq experiment (data shown in Supplementary Fig. 2A, B), we sorted 1000 WM989 primed cells (based on NT5E expression) and 1000 mixed WM989 cells into one well of a 96-well plate, and 2000 WM989B cells into another well. We then allowed the cells to undergo 4 divisions, trypsinized them, and prepared the samples all the way through library generation as described in the Chromium Single Cell 3′ Reagent Kit V3.1 (Dual Index) user guide.

For the third scRNA-seq experiment, shown in Fig. 5A, we sorted 2000 barcoded WM989 cells into a single well of a 96-well plate. We waited for these cells to expand through 7–8 divisions, and then randomly split these cells across four separate plates. We waited one day for the cells to adhere to the plate, and then started treatments (one plate untreated, one plate 5 ng/mL TGFB1, one plate 4 μM LY2109761 (TGFBRi), and one plate 2μM GDC-0941 (PI3K inhibitor)). We incubated the cells in their respective treatments for 5 days. We carried the above steps with two samples in parallel as replicates. After 5 days, we trypsinized the cells and processed them all the way through library generation as described in the chromium Single cell 3′ Reagent kit V3.1 (Dual Index) user guide.

We sequenced all our single-cell libraries using a NextSeq 500 with the High Output Kit v2.5 (75 cycles, Illumina, 20024906). For samples sequenced with the Single Cell 3′ Reagent Kit V3 (single index), we used 8 reads for the index, 28 reads for read 1, and 49 reads for read 2. For samples sequenced with the Single Cell 3′ Reagent Kit V3.1 (dual index), we used 10 cycles for each index, 28 cycles for read 1, and 43 cycles for read 2.

### Lineage barcode recovery from scRNA-seq

To recover the lineage barcodes, we used an aliquot of the excess full-length cDNA generated in the 10x library protocol. Specifically, we selectively amplified reads containing the lineage barcode using primers that flank the 10x cell barcode and the end of the lineage barcode in our library (Supplementary Table 1)[69]. To perform the PCR, we combined 100ng of full-length cDNA per reaction, 0.5μM of each primer, and PCR master mix (NEB, M0543S). We used 12 cycles to amplify the cDNA using the following protocol: an initial 30 s denature step at 98 °C, then 98 °C for 10 s followed by 65 °C for 2 min repeated 12 times, and a 5-min final extension step at 65 °C. We then extract the amplified barcodes, which are ~1.3 kb, using SPRI beads (Beckman Coulter, B23317) for size selection (0.6X bead concentration). To sequence the barcode library, we used a NextSeq 500 with a Mid Output Kit v2.5 (150 cycles, Illumina, 20024904). We performed paired-end sequencing and used 28 cycles on read 1 to read the 10x barcode and UMI, 8 cycles on each index, and 123 cycles on read 2 to sequence the lineage barcode.

### gDNA barcode recovery

To sequence barcodes from gDNA, we trypsinized cells, pelleted them, and then extracted their gDNA using the QIAamp DNA Mini kit according to the manufacturer's protocol (Qiagen, 56304). To amplify the barcodes, we performed PCR using primers with homology to each side of the barcode. The primers also contain the Illumina adapter sequence, and index sequences (see Supplementary Table 1 for primer sequences). To perform the PCR amplification, we used 500ng isolated gDNA, 0.5 μM of each primer, and PCR Master Mix (NEB, M0543S) for each reaction. We used 24 cycles to amplify the barcodes using the following protocol: an initial 30-s denature step at 98 °C, then 98 °C for 10 s followed by 65 °C for 40 s repeated 24 times, and a 5-min final extension step at 65 °C. After amplification, we used SPRI beads (Beckman Coulter, B23317) to select the amplified barcode product (expected length of ~350 bp). To isolate this fragment size, we performed a two-sided selection where we first selected with a 0.6× bead concentration and kept the supernatant (large gDNA fragments were on beads). We then select again using a 1.2× bead concentration keeping the material bound to the beads(small fragments such as the primers were in the supernatant). To sequence the barcode library, we

used a NextSeq 500 with a Mid Output kit (150 cycles, Illumina, 20024904). We performed single-end sequencing and used 151 cycles on read 1 to read the lineage barcode and 8 cycles on each index.

## scRNA-seq analysis

We used the 10x Genomics Cell Ranger pipeline to generate FASTQ files (using the hg38 reference genome), to assemble the count matrix, and to aggregate replicate runs (without depth normalization). We also used the Cell Ranger feature barcode pipeline to integrate our lineage barcodes with the scRNA-seq data (more information in the section titled Combining single RNA sequencing and barcode data).

Once we generated the aggregated count matrices with incorporated barcode information, we analyzed the data using Seurat V3[70]. Using Seurat, we performed basic filtering of the data based on the number of unique genes detected per cell, both removing poorly sequenced cells (low number of genes), and data points likely to be doublets (high number of genes). We also filtered based on the percent of mitochondrial reads to eliminate low-quality or dying cells. If we saw batch effects between replicates, we used the Seurat scRNA-seq integration pipeline to remove them. When there were no batch effects, we used SCtransform to normalize the data before running PCA, clustering, and dimensionality reduction with UMAP. When plotting gene expression information, we did not use the SCtransform data, but rather separately log normalized the data. To generate single-cell signature scores for a gene set, we used the UCell package[71]. We selected the primed cell gene set by including all genes with a positive $\log_2$ fold change in our list of differentially expressed genes between primed and drug-susceptible cells (Supplementary Data 1).

## Combining scRNA-seq and barcode data

To identify the lineage barcodes from the sequencing data, we used a custom python script (available through GitHub here: https://github.com/SydShafferLab/BarcodeAnalysis) and the 10x Genomics Cell Ranger Feature Barcode pipeline. In this pipeline, we first identified lineage barcodes in the FASTQ files by searching for a known sequence at the beginning of all lineage barcodes. Once we identified all potential barcode sequences, we used the STARCODE[72] to identify barcodes that were very similar to each other and replace them all with the most frequently detected sequence within the set of similar barcodes. We then put these modified barcode sequences back into the FASTQ file and generated a reference file containing all the edited barcode sequences. Next, we fed these edited FASTQ files and the reference file into the Cell Ranger pipeline and used the Feature Barcode analysis function to link lineage barcodes with the cell barcodes. This provided us with the lineage and gene expression information for cells where a barcode was identified.

Our initial steps identified barcodes by combining similar barcodes, but when we looked at this output we found that we could more stringently call real lineages using additional filtering steps. The code used to accomplish this can be found in the Assign a lineage to each cell section of the 10X1_r1_r2_Analysis_unorm_sctrans.Rmd script available on the Google Drive link in the Software and data availability section. In brief, we first eliminated lineages that appear across multiple samples, as such lineages are not possible. We then also removed lineages that are bigger than expected given the amount of time cells were given to proliferate. Finally, for cells that appeared to have more than one lineage barcode, we tested whether there are multiple cells with this same combination of barcodes and considered those cells to be in the same lineage.

## Validating primed cell markers

To test whether different proteins are markers of the primed cell state (NT5E, NGFR, EGFR), we stained live WM989 cells with an antibody for the marker of interest and then sorted the stained cells. Specifically, we sorted a mixed population of cells in one well, and a population of cells

high in our marker of interest in another well (for sorting detail refer to the Fluorescence-Activated Cell Sorting section). We then allowed the cells to adhere to the plate for 24 h and then started treatment with 1μM vemurafenib for 3 weeks. After 3 weeks in vemurafenib, we counted the number of cells and drug-resistant colonies in each well to determine if the marker increased the number of cells that survive targeted therapy. The percentage of cells sorted in the primed condition was determined by sorting different percentages of high cells and treating them with targeted therapy to identify which percentages were resistant.

## ATAC-seq

We sorted 10,000 cell populations of EGFR/NGFR-High, EGFR-High, NGFR-High, and negative (for both markers) cells in triplicate as described in the FACS section of methods. Immediately after sorting, we performed OMNI-ATAC on each population of cells[73]. We used the Illumina Tagment DNA Enzyme for tagmentation (Illumina 20034197) for tagmentation and performed two-sided bead purification before sequencing. We performed paired-end, single-index sequencing on pooled libraries using a 75-cycle NextSeq 500/550 High Output Kit v2.5 (20024906) allotting 38 cycles to both read 1 and read 2 and 8 cycles to the sample indices.

## ATAC-seq alignment and analysis

We adapted the paired-end analysis pipeline from ref. 74 for alignment, processing, and peak calling. Briefly, we aligned reads to hg38 using bowtie2 v2.3.4.1, filtered out low-quality read alignments using samtools v1.1, removed duplicated reads with picard 1.96, and generated alignment files with inferred Tn5 insertions. To call peaks, we used MACS2 2.1.1.20160309. We then identified consensus peaks using the findConsensusPeakRegions function in the consensusSeekeR package in R as peaks seen in at least 3 replicates out of 12 total[75]. We then counted reads within these consensus regions for each sample and created a DESeq2 object which we used to perform PCA on consensus peaks[76]. We then plotted a row-scaled heatmap with ward.D2 clustering of the top 20,000 most variable peaks.

## Mouse model tumor generation

The PDX tissue was collected by the Weeraratna lab as part of a prior study[6]. Briefly, these melanoma tumors were generated by subcutaneously injecting $1 \times 10^6$ WM989-A6-G3-Cas9-5a3 cells into 8-week-old NOD/SCID mice. The mice were fed AIN-76A chow, and the facilities were maintained between 21–23 °C, a humidity of 30–35%, and lights had a 12 h on/off cycle with lights on from 6:00 to 18:00. The tumor was collected when it measured ~1500 mm³. Tumor blocks were embedded in OCT, flash frozen, and stored at −80 °C.

## Tissue RNA FISH

To analyze NT5E and SOX10 expression in mouse tumors, we used HCRv3.0 with probes targeting NT5E and SOX10 [25,77]. The probes and fluorescently labeled hairpins were purchased from Molecular Instruments (NT5E lot #: PRK825, SOX10 lot #: PRK826). To perform HCR in tissue, we made slight modifications to published protocols[25,77]. First, we used cryostat sectioning to generate 6μm sections of the fresh frozen tumor. We placed these sections on charged slides and fixed them with 4% formaldehyde for 10 min. We then washed the slides twice with 1X PBS and stored them in ethanol.

To start HCR, we placed the slide in a slide staining tray and washed the slides twice with 5× SSC (sodium chloride sodium citrate). After removing the 5× SSC, we added 200 μl of hybridization buffer (30% formamide, 5× SSC, 9mM citric acid (pH 6.0), 0.1% Tween 20, 50 μg/mL heparin, 1X Denhardt's solution, 10% dextran sulfate) which was pre-heated to 37 °C onto the tissue. We then incubated the slide for 10 min at 37 °C. All incubation steps in this protocol were done with the slide staining tray closed and with water at the bottom of the tray to

prevent the sample from drying out. During this incubation period, we added 0.8 pmol of each probe pool (in this case *NT5E* and *SOX10*) to 200 μl of probe hybridization buffer pre-heated at 37 °C and kept the solution at 37 °C. After 10 min, we removed the hybridization buffer from the tissue and added 300 μl of hybridization buffer containing the probe pools. We placed a cover slip over the sample and incubated it for 12–16 h at 37 °C. After the incubation, we prepared the hairpins by putting 0.6 pmol of each hairpin into its own tube (to keep hairpin 1 and hairpin 2 separate), and then performed snap cooling by heating them to 95 °C for 90 s and then slowly cooled down to 25 °C over 30 min in a thermocycler. While the hairpins snap cooled, we added 300 μl of wash buffer (30% formamide, 5× SSC, 9mM citric acid (pH 6.0), 0.1% Tween 20, 50ug/mL heparin) to the slide to remove the cover slip. We then performed multiple wash steps with decreasing amounts of wash buffer in the solution. We first added 300 μl of 75% wash buffer, 25% 5× SSCT (5× SSC with 0.1% Tween 20), removed that and added 300 μl of 50% wash buffer, 50% 5× SSCT, removed that, and added 25% wash buffer, 75% 5X SSCT, and finally removed that and added 300 μl of 100% 5× SSCT. For each step of the wash, we left the slides in solution for 15 min at 37 °C. After the last wash, we removed the 5× SSCT and added 200 μl of room temperature amplification buffer (5× SSC, 0.1% Tween 20, 10% dextran sulfate) to the slide and incubated it at room temperature for 30 min. We then removed the amplification buffer and added the prepared hairpins mixed in 100 μl of amplification buffer to the slide, and added a cover slip. We incubated the slide in the staining tray at room temperature for 12–16 h. After incubating with the hairpins, we removed the coverslip and washed the slide off using successive 5× SSC washes. We put 300 μl of 5× SSC on the sample for 5 min, removed it, then added 5× SCC for 15 min, removed it, added 5× SSC for 15 min again, removed it, and finally added 5× SSC with DAPI for 5 min. After the washes, we mounted the slide using TrueVIEW (Vector labs, SP-8500-15), added a coverslip, and sealed it with nail polish.

## Flow cytometry

We dissociated cells from the plate using trypsin-EDTA into a single-cell suspension and washed once with 0.1% BSA. We then resuspended the cells in a 1:200 dilution of anti-NT5E antibody conjugated with APC and incubated them for 30 min on ice. Next, we washed the cells once with 0.1% BSA, once with 1% BSA, and then resuspended them in 1% BSA for analysis by flow cytometry. We used an Accuri C6 for our flow cytometry and quantified 10,000 events per sample. To analyze the data we used the R package flowCore[78]. In our analysis, we used forward and side scatter to identify cells, and used the FL4 channel (640nm excitation laser and 675/25 filter) to quantify cell surface levels of NT5E. To determine what percent of cells were primed, we set an intensity threshold where 2% of untreated cells would land above the threshold. We considered any cells above this threshold as primed.

## Drug-resistant colony experiments

We plated cells in six-well plates with 10,000 cells per well. After plating, we gave cells 24 h without treatment to adhere to the plate. We then initiated pretreatments and changed the media on the no-pretreatment controls. During the pretreatment period, we treated the cells with doses of the drug that had low toxicity and minimal effect on the proliferation rate of the cells (assay for determining the doses described in the Pretreatment growth effects section). We incubated cells in their respective pretreatment for 5 days. For experiments where we normalized for drug resistance, at this point 3 wells were fixed and the number of cells quantified to determine how many cells were present in each condition after pretreatment. We then aspirated the media and replaced it with media containing 250 nM dabrafenib and 2.5 nM trametinib. We maintained treatment with 250 nM dabrafenib and 2.5 nM trametinib for 4 weeks, changing the media every 3–4 days. After 4 weeks, we fixed the cells by aspirating off the media,

washing the wells with DPBS, and treating them with 4% formaldehyde for 10 min. We then aspirated off the formaldehyde and washed twice with DBPS. Finally, we added 2 mL of DPBS to each well and stained the cells with DAPI. We then imaged the wells using a 10× objective on a fluorescence microscope (Nikon, Eclipse Ti2).

## Cell and colony counting

To determine how many cells were in wells after drug treatment, we used a custom pipeline called DeepTile (https://github.com/arjunrajlaboratory/DeepTile/tree/071e3e9fb27f50ce024fd5ece25e3a4b0071f771) to feed tiled images into DeepCell to generate nuclear masks[79,80]. To simplify the interface with DeepTile and DeepCell, as well as remove nuclei incorrectly called outside the well, we used a custom tool DeepCellHelper (https://github.com/SydShafferLab/DeepCellHelper). We then determined the number of cells per well by counting the number of masks per image. We also developed our own machine-learning approach to identifying nuclei called NucID which was 10X faster (https://github.com/gharmange/NucID). This was used for counting nuclei in the pretreatment normalization of drug resistance experiment. To identify colonies, we used a custom graphic user interface ColonySelector (https://github.com/SydShafferLab/ColonySelector) to circle individual colonies in each well and save a file containing which nucleus belongs to which colony. Using the output of the colony selecting software, we counted the number of colonies there were in each well.

## IncuCyte imaging and analysis

For time-lapse experiments on the IncuCyte S3 (Sartorius), we used a clonal population of WM989 cells tagged with H2B-GFP for nuclear tracking. We took 4× images with a 300 ms exposure for GFP every 12 h to track cell growth over time. We used the IncuCyte software to generate nuclear masks and exported csv tables containing the number of nuclei in each well at each time point. We analyzed this data in R.

## Pretreatment growth effects

To determine if our treatments were leading to state-specific changes in proliferation rates, we used a clonal WM989 H2B-GFP tagged cell line. To isolate drug-susceptible and primed cells, we sorted on NT5E and separated drug-susceptible and primed cells into separate wells. After 24 h to adhere to the plate, we added TGFB1, PI3Ki, or nothing to the media. We then imaged the cells according to the description in the Incucyte imaging and analysis section.

## Western blot analysis

WM989 melanoma cells were sorted either as mixed (all live cells) or as primed (top 2% of NT5E expressing cells). Cells were allowed to grow for 24 h before the initiation of treatment. PI3Ki treated cells had PI3Ki (2uM) put on them for 4 h before harvest. Starved cells had media without FBS. Cells treated with insulin growth factor (IGF) had 200 ng/ml IGF added to them 15 min before harvest. For combined treatment with PI3Ki and IGF, cells were treated with PI3Ki for 3 h and 35 min and then treated with PI3Ki and IGF for the last 15 min before harvest. Cells were harvested in RIPA buffer (50 mM Tris pH 8.0, 150 mM NaCl, 0.5% NP-40, 0.1% Sodium Deoxycholate, 0.1% SDS) containing Halt Protease and Phosphatase Inhibitor Cocktail (Thermo Fisher Scientific, Cat. 78445) and centrifuged at $10,000 \times g$ for 10 min at 4 °C. Protein concentrations were determined by the Pierce BCA protein assay kit (Thermo Fisher Scientific, Cat. 23225). Protein lysates (10–20 μg) were resolved by Tris-Glycine SDS-PAGE and transferred to nitrocellulose membranes (Bio-Rad, Cat. 162-0115, 0.45 mm pore size for all experiments). All membranes were incubated with the indicated primary antibodies overnight at 4 °C and were diluted in TBST (20 mM Tris pH 7.5, 150 mM NaCl, 0.1% Tween-20) supplemented with 5% bovine serum albumin (BSA, Sigma-Aldrich, Cat. A7906). The following antibodies were used: Phospho-AKT Ser473 (1:1000, Cell Signaling

Technology #4060), pan-AKT (1:1000, Cell Signaling Technology #9272), β-actin (1:1000, Santa Cruz Biotechnology #4778). Primary antibodies were detected with horseradish peroxidase-conjugated secondary antibodies followed by exposure to ECL reagents (Perkin Elmer, Cat. NEL105001EA).

## Reporting summary
Further information on research design is available in the Nature Portfolio Reporting Summary linked to this article.

## Data availability
The scRNA-seq data generated in this study have been deposited in Gene Expression Omnibus and is publicly available under accession code GSE237228. The publicly available data used in this study is available in the GEO database under accession code GSE115978, GSE77940[27]. The remaining data are available within the Article, Supplementary Information or Source Data file. Source data are provided with this paper.

## Code availability
All analysis code and accompanying data used for this paper is available at the following link: https://drive.google.com/drive/folders/1-C78090Z43w5kGb1ZW8pXgysjha35jlU?usp=sharing. The following custom pipelines used in the paper are available on github: NucID: https://github.com/gharmange/NucID. ColonySelector: https://github.com/SydShafferLab/ColonySelector. BarcodeAnalysis: https://github.com/SydShafferLab/BarcodeAnalysis.

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

## Acknowledgements

We thank all members of the Shaffer lab for feedback on experiments and the manuscript, P. Gonzalez-Camara and T. Ridky for thoughtful discussions and ideas, L. Bugaj for feedback on the manuscript, and M. Herlyn for providing cell lines. S.M.S. acknowledges support from the NIH Director's Early Independence Award DP5OD028144 and the Wistar/Penn Skin Cancer SPORE, P50 CA261608. A.T.W. acknowledges support from R01CA207935 and P01CA114046. M.E.F. acknowledges support from R00CA263017. L.C.K. acknowledges support from T32 CA09140 from the NIH NCI. R.A.R.H. acknowledges support from the NSF

Graduate Research Fellowship (DGE-1845298). A.S. acknowledges support from NIGMS (NIH) under award number R35GM148351.

## Author contributions

G.H. and S.M.S. conceptualized the project and designed the study. G.H. performed all experiments and analyses with the following exceptions. R.A.R.H. helped with the barcode processing pipeline and provided helpful discussion. D.S. performed the ATAC sequencing experiment and its analysis and helped with the barcode processing pipeline. M.S. and A.S. performed the modeling analysis. B.E. provided technical guidance and troubleshooting for the barcoding library. C.S. and L.K. measured and analyzed the activity of signaling pathways. A.T.W., M.E.F., and G.M.A. generated the mouse PDX tissue. Z.N. developed parts of the image analysis pipeline. S.N. helped maintain cell lines and performed computational analyses. G.H. and S.M.S. wrote the paper.

## Competing interests

The authors declare no competing interests.
