## [Peer Review file · Nature Communications]

REVIEWER COMMENTS

Reviewer #1 (Remarks to the Author): expertise in single cell RNA-seq methodology

This manuscript by Harmange et al. details a series of experiments using single-cell lineage tracing on melanoma cancer cell lines to delineate plasticity-based mechanisms of drug resistance. Overall I think the study could have great impact in the field of cancer biology, with deep implications for understanding mechanistically how population dynamics within tumor cells are orchestrated. Particularly, I find very interesting the identification of a set of drug-susceptible cells that are capable of transitioning to a resistance-primed state. Although I believe these results are interesting, the manuscript is a bit poorly organized. More than a few times I found it troubling to identify the conclusions to certain results or I had to piece them together from other multiple parts of the main text or the discussion. Other times, I felt like the authors jumped to conclusions without clearly describing the results in the main text (even if the figure and experimental design schematics are really helpful to make it cohesive). Altogether, this is not so much a problem with the data/conclusions, but rather the organization and presentation of results. Otherwise, my major concern relates to the supposed identification of "intermediate state" cells. Several data suggest to me that this is not truly an intermediate population. First, authors do not find clear evidences of intermediate states by UMAP or trajectory analysis (although this could be done in further depth). Second, the intermediate states appear to be present (at some rate) in both sorted samples, so defining the origin of these supposedly bi-fate clones in a definitive fashion seems difficult. Third, this population is never isolated/enriched and tested independently to validate their behavior. Fourth, it is not clear to me whether these clones that appear in both "susceptible" and "primed" regions are ever found outside these intermediate states, indistinguishable from other fully susceptible cells. The authors do identify enrichment of EMT/TGFbeta signaling pathway in them and then they use activators to confirm that susceptible-cells can be turned into primed cells by simple addition of TGFb. This last series of experiments, together with the state-switching stochastic model is, to me, the best part of the paper. However, I remain unconvinced that the intermediate state is not in fact a relatively long-lived, clonally-imprinted population of "bipotent" cells. In sum, I believe this is a very good study that will have great impact in the field, but some issues need addressing before publication. My point-by-point comments follow below:

In the description of the first experiment, at the beginning of the main text, lines 95-117 describe in detail a single-cell lineage tracing to be performed (Fig 1A-C). But then, 119-127 talk about the single-cell clustering results obtained (Fig. 1D). And then lines 129-147 introduce experiments and results with PI3K inhibitors (Fig. 1E-G), and they go on and on, without ever mentioning the lineage tracing data. And, as a reader, I get completely lost. I think these experiments might be a bit out of place in this section. I would perhaps first introduce the single cell data results clearly, then the changes with inhibitors, (Fig 1D-G) and then start with describing methodologically the lineage tracing approach in the next figure (Fig 2). Also, I don't understand how these results (PI3Ki data) fit together at this point in the manuscript. They seem a bit out of place, and perhaps better introduced before the data from Figure 5?

If NT5E is a better marker to homogeneously enrich resistant cells, it should be validated against NGFR-high and/or AXL-high cells instead of just to a mixed control (in Supp Fig 1F-G).

The lack of transitioning cells between the susceptible and primed cluster is interesting, but alternative explanations are not sufficiently tested or discussed. First of all UMAP visualization can be misleading, depending on the PCAs selected. Cell cycle score regression was not performed, and this may be forcing the separation of the few "intermediate" cells around the plot, preventing identification of a more clear transition bridge between susceptible-primed groupings. Also, different representations, such as diffusion component analysis or force-directed layouts might be tested. These are sometimes better at representing distances between the different transitioning cells. Do the authors suggest

that increasing the number of sampled cells would identify the trajectory properly? Or is it also possible that the susceptible-to-primed state transition is not gradual or continuous?

In line 174, how is a "primed cell" and a "drug-susceptible" cell defined? This is not clearly described in main text, legends or methods. What I got from reading the manuscript is that they are classified by some form of cluster annotation. Previous publications from these authors have shown that these marker-based clusters (AXL^{high}, EGFR^{high}) are "enriched" for resistant states, but don't purify them to homogeneity. Wouldn't it be more stringent to classify them by a real drug-treatment response in a split of the clones? Is anything preventing this?

The state-switch models used throughout are really useful. However, the first introduction to them is a bit confusing (compared with their later implementation in lines 372-395, where they are more clearly explained). As an example, in line 184, right after discussing the differential exp analysis, the reader is obliged to remember what "from this data" means. And it doesn't actually mean the differential expression analysis discussed in lines 180-183 (just before), it actually means the clonal data analysis from lines 171-179, or does it? This is confusing to me.

Many terms are used to define the transitioning drug-susceptible cells. Intermediate state, intermediate cells, crossing-lineage. Some unified terms would help in understanding what is being discussed. In line 211, it is said that the "intermediate cells" would have not been identified using standard scRNAseq techniques. Some evidence of this would be helpful to strengthen this statement (for instance, subclustering, PCA and reclustering analysis of the drug-susceptible clusters), especially since 575 genes are differentially expressed between these and non-transitioning drug-susceptible cells.

Determining the initial cell state of the labeled clones by sort-enrichment is probably insufficient and single-cell RNAseq + clone splitting (as in Figure 5) would have been a better way of defining the initial states of cells and their fate-switching. Instead, two different populations were sorted and separately cultured and profiled some time after barcode labeling. But what were these populations really initially (before so many doublings)? If one population was primed-enriched and the other one contains a mix of both susceptible and primed-resistant states, then the origin of the bi-state clones could be misinterpreted. Also, the label on Fig 1C seems misleading, because it suggests that drug-susceptible cells were sort-enriched, and not a "mixed population" as stated in both the figure legend and text. The switching from primed-to-susceptible is clear, but the switch from susceptible-to-primed is not as clear to me if the sorted cells were not relatively pure at starting point. If this is all an issue of AXL/NGFR/EGFR expression heterogeneity, then perhaps NT5E could be used to purify the susceptible cells much better?

More worryingly, EGFR, and especially AXL, expression seem to be slightly enriched partially in the bottom right of the susceptible clusters, very close to the location of some of the intermediate/switching states. In fact, in Supp Figure 1B, it seems that those intermediate states are even enriched in the EGFR/NGFR^{high} sample, though maybe a bit hidden by the layering. It would be great to visualize the colors with some transparency, or generate separate plots/quantifications of cluster enrichment for each sample to improve visualization. This all suggests that perhaps "intermediate state" are actually deterministic, heritable, clonally-imprinted bi-potent states, closer to the drug-susceptible region, that are deterministically giving rise to both susceptible and primed states over time. This could be especially concerning if this population is present in both of the sorted samples.

Do intermediate cells predictably become drug resistant with a higher rate than non-intermediate drug-susceptible cells? Isolating these cells would be really important to validate these results. Perhaps, if I am correct, then NT5E^{low} AXL/EGFR⁺ cells would be a reasonable suggestion to enrich them, since it appears that AXL/EGFR are high at least in some regions of the intermediate state, and NT5E is solely expressed at high levels in the primed clusters.

Is there a continuous trajectory of gene expression state changes throughout intermediate cells that leads to primed cells? Or do the authors support the idea that this is rather a stochastic gene expression state connected to a higher probability of transition to the resistance-primed state?

Expression scale in Fig 2J is missing.

The drug-induced fate-switching experiment in Figure 5 is really cool. A second untreated sample would have been useful to quantify the baseline rate of lineage conversion of those clones. Maybe the TGFBRi-treated sample serves that purpose to some extent, since there seems to be little to no effect with this compound.

In the clone-splitting experiment, now, for the first time I find the most convincing data regarding the primed-to-susceptible switch. Clearly this is only occurring for a handful of clones here, so it's difficult to assess... however is it possible to compare this data with the bipotent fate data from figure 2 and see the differences? I fear that clonally imprinted AXL/EGFRmid/+ states within the NT5E-low subpopulation of susceptible cells might be at the origin of the bipotent behaviors observed, here identified as "intermediate states". This alternative interpretation needs to be challenged properly. Do switching cells ever produce proper drug-susceptible cells that are indistinguishable from non-switching cells?

In Figure 5G, the response of clones to TGFβ1 is super heterogeneous. Are there any (transcriptional) differences between the clones that "respond" to TGFβ1 and become primed and the clones that do not? Are they connected to the expression signature of the intermediate state? It would really increase the impact of the manuscript if specific signatures within susceptible cells are present and predict their TGFβ1 response.

Reviewer #3 (Remarks to the Author): expert in single cell transcriptomics bioinformatics

The authors identify a rare population of melanoma cells which is primed for resistance to treatment from a BRAF and MEK inhibitor combination. The authors characterize the kinetics of the transition between the susceptible and resistant subpopulations using scMemorySeq. They also identify key markers such as NT5E and the EMT-like state to define the primed state or cells transitioning to the primed state. They go on to show that inhibiting PI3K signaling and supplementing TGFβ1 ligand can drive cells to switch to susceptible or primed states, respectively. Concerns with this paper relate to the soundness of their conclusion about cellular memory which is critical to the novelty of the work since the presence of such a pathway to cause resistance to BRAF/MEK inhibitor has been well trodden (Smith et al Cancer Cell 2016), and evidence that a subpopulation is responsible has been described (Fallahi-Sichani Mol. Bio.Sys 2017). In the absence of convincing and clear rationale for the concept of cellular memory, it is unclear what else is being added here in the absence of any deep biological or pharmacologic investigation.

Major Comments

Related to Fig. 1E and Supplementary Fig. 3A-C, the authors indicate that by modulating the number of primed or susceptible cells they can alter the number of eventual resistant cells. However, Supplementary Fig3D shows that different treatment strategies tested can result in differences in cell doublings; these differences may affect endpoint quantification of the number of resistant cells and/or colonies. The authors should show supporting proliferation and apoptotic data of their pretreatment strategies or normalize data to pretreatment controls.

For example in Figure 1E, the authors claim that the dose used for pretreatment with the PI3K inhibitor did not kill cells nor did it impact cell proliferation; however, in supplementary figure 3D, pretreatment with PI3K is shown to result in decreased cell doublings. The authors should show their analysis for cell proliferation and apoptosis upon PI3K inhibition and show how they derived at their optimal concentration.

The authors claim that the primed and susceptible cell states are two discrete states as shown in the two clusters seen in Fig. 1D, however in 5B there does not seem to be a similar discretization in the untreated cells. Moreover, the switch between primed and susceptible seems to be a continuous phenomenon evidenced by the distribution of NT5E expression in the population as shown in Fig. 4B and 5B. Thus it would appear that cells are undulating through a continuum instead of jumping between discrete configurations of those pathways and that some drugs are altering the level of this specific gene signature. Could an alternative model be that NT5E/EMT level in all cells is turned up or down by TGF β or PI3K inhibition, leading to some cells crossing a threshold, making it appear that there is a switch in cell state? What if the histograms in 4B are overlaid, do the drugs inhibit or enhance levels of NT5E in the majority of cells (i.e. shift in median level of NT5E)?

In Fig. 2D, MITF is downregulated in sister cells of primed cells compared to lineages without primed cells. Also, in Fig. 2J the primed vs susceptible heatmap shows that MITF is further downregulated in primed cells. It is very well established in the literature that BRAF/MEKi resistant cells in melanoma upregulate MITF (Smith et al, 2016 Cancer Cell, Johannssen Nature 2013 among others). What is the reason for this discrepancy, is there a process where the full resistant cells go from downregulating MITF to then upregulating it?

Minor comments:

The NT5E high cutoff for untreated cells used in Fig. 4B should be changed such that the ratio of primed to susceptible cells is the same as shown in the two clusters in both Fig. 1D and Fig 2B for consistency. Ideally the cutoff should be based on the percent of cells that survive BRAFi/MEKi treatment alone.

In the case of phenotype switching of barcodes from primed to susceptible and vice versa (Fig. 5G and 5H), the fraction of cells from the respective lineages in the primed state for the untreated sample should be shown for better readability of data.

In Fig. 3A, the authors should quantify the expression of SOX10 and NT5E in the region of interest compared to a control region.

Fig. 3B and Fig 3D require controls (normal tissue or untransformed samples) for primed and susceptible states in order to show that the heterogeneity captures something outside the normal range of expression.

In Fig. 5G, a better visualization of the concept shown would be to take a representative number of lineages (say 20 lineages) and connect them via a line across the different treatments. This would show how each specific lineage changes in the fraction of primed cells by treatment.

The authors should show that they can recapitulate the two clusters observed in Fig. 1D UMAP from the untreated cells in Fig. 5B,C,D to better define the primed and susceptible

populations. Can you take the annotations from this newly created UMAP and map them onto the UMAP shown in Fig. 5B,C,D to see if it aligns with your definition of primed and susceptible populations shown in that figure?

Given that primed and susceptible cells have different growth rates, it may be reasonable to assume that there might be differences in the cell cycle which could explain the clusters in Fig 1D. The data should have cell cycle markers regressed out. Supplementary data Fig. 2B shows a high difference in % of cells in S phase between the two clusters. If the two clusters no longer remain after regressing out those markers, then there is a significant flaw in using this method to call cells as primed or susceptible.

Reviewer #4 (Remarks to the Author): expert in RNA biology and transcriptional regulation

Summary:

In this interesting study, Harmange and colleagues developed a single-cell RNA-sequencing based approach to measure cell state memory in mammalian cells. This is of clinical relevance as it has been previously shown that melanoma cells can switch between therapy resistance (primed) and drug-naïve cell states. To measure cellular memory the authors aimed to track transcriptional states across cell lineages. Towards this, Harmange et al applied a barcoding library to melanoma cells and subsequently profiled the transcriptome of those barcoded cells at a single-cell level after a certain number of cell doublings. Through lineage tracing with cell barcoding the authors found that most cells maintained their cell state memory, which was in contrast to their previous conclusions from bulk RNA-sequencing studies that indicated dynamic fluctuation between therapy resistant and drug-naïve gene expression states in the same studied cell line. However, in their new manuscript using single-cell analysis the authors identified a pool of cells that lost their gene expression memory and they demonstrated that this lineage switching is dependent on PI3K and can be induced by TGFB1. They also confirmed that similar gene expression signatures exist in mouse models and patient samples. Finally, the authors demonstrated that the transcriptional memory remains stable over time and that treatment with TGFB1 and PI3Ki induces primed and drug-susceptible states, respectively. Although only a small fraction of cells switches states upon those treatments this is still sufficient to alter the outcome of treatment regimens across a cell population. Overall, the experimental approach (scMemorySeq) and conclusions of this study are novel, accurate, clinically relevant and of great interest to a broad readership. The experiments are well controlled, data are presented in a clear and visually attractive manner and methods are described in sufficient details. Overall, this manuscript is suitable for publication after addressing the minor comments listed below.

Minor comments:

- Line 50-53: It would be helpful if the authors could summarize their previous work in a bit more detail. Maybe they could add 1-2 sentences briefly describing the technology and cancer type used along with major finding from the cited studies.
- Line 242: delete "population"
- Line 335: Shouldn't this read "SOX-high" instead of "SOX-low"?
- The authors apply a PI3K inhibitor and show that this decreases drug resistance in melanoma cells. PI3K inhibition should ideally be confirmed by assessing phosphorylation state of kinase targets by western blotting.
- Also, can the authors comment on whether addition of TGFB1 induce any markers of EMT as observed in intermediate cell states (using scRNA-seq data from Fig. 5)?
- Given the prominent association with primed cell state, the authors should consider displaying the NT5E marker gene in the heat maps displayed in Fig. 5F. Do the NT5E transcript levels correlate with the protein level shown in Fig. 4C?

Reviewer #5 (Remarks to the Author): expertise in melanoma drug resistance

In their manuscript, Harmange et al present a novel method for in vitro tracking of phenotypic switching using barcoding and scRNAseq. Using this method, they explore the kinetics and dynamics of cellular switching between a drug-sensitive state and a state primed for drug resistance in melanoma. Authors then examine how pre-treating cells with inhibitors of TGFBR and PI3K based on the transcriptional characteristics of resistance "primed" states affect these transitions. These data provide insights into the molecular pathways which may regulate the transitions between cell states. Data gathered in these studies was then utilized to parameterize a mathematical model of phenotypic switching between the drug-susceptible and drug-primed state. Although many of the transcriptional characteristics of the drug-sensitive and drug-resistant states have been defined previously, the kinetics and dynamics of cell transitions through these states have never been explored and are of high interest to the melanoma research community. This work also provides a deeper functional dimension to some of the pathways previously implicated in BRAF inhibitor resistance.

Questions/concerns:

- 1.** Is it possible that melanoma tumors have more than 1 drug resistance "primed" and more than 1 drug-sensitive state? There have been a few papers now that have identified 46 transcriptional states in melanoma, most recently a paper from the Marine lab in Nature volume 610, pages 190–198 (2022). Is there any indication how the drug sensitive and drug "primed" states defined in this manuscript fit within the 6 cellular state landscape described in the Marine lab study? It would be helpful to understand how the states identified here fit into the published states. For example, is the "primed" state the same cells that are identified as "mesenchymal-like"? and are the sensitive states a combination of all others? Or are mesenchymal-like the very rare subset of the "primed" state that are then also resistant to PI3Ki pre-treatment coupled with BRAF inhibitor treatment?
- 2.** Is pre-treatment with PI3K more, less, or similarly as effective compared to simultaneous combination treatment? This is an essential question that needs to be answered and would have a large impact on how dynamic treatment strategies are developed preclinically and clinically.
- 3.** Importantly, the authors show that pre-treating cells with PI3K significantly reduces but does not completely abolish the outgrowth of resistant cells. The authors should analyze the transcriptional profiles of rare cells in the lineage which did not respond to PI3Ki. It is essential to know more about these states, how they are different from BRAFi-sensitive and "primed" states, and potential insights into how they can be targeted.
- 4.** If the drug-susceptible cells in lineages with a high proportion of cells already in the primed state are more easily switched into the primed state, is it possible to predict clinical response to BRAFi based on the content of the primed state in a tumor?
- 5.** The authors are tackling a phenomenon that has been studied intensely for the last decade. It is important to be more thorough about setting the context for this study and make sure cite additional papers related to:
 - a)** Reversal to drug-sensitive state and leveraging of state dynamics in therapy scheduling
 - b)** AXL/MITF/SOX10/FN1 transcriptional states related to BRAFi resistance
 - c)** Role of TGFbeta in BRAFi response
 - d)** Role of PI3K inhibitors in reversing BRAFi resistanceAlthough authors do cite some work in some of these areas, it could be more thorough.

Minor questions/concerns:

- 1.** Why did almost half the cells profiled not have barcodes (related to results for Figure 1)? Are barcodes lost over time or were barcoded and non-barcoded cells analyzed?

2. There is a typo on line 24

Reviewer #1: expertise in single cell RNA-seq methodology

This manuscript by Harmange et al. details a series of experiments using single-cell lineage tracing on melanoma cancer cell lines to delineate plasticity-based mechanisms of drug resistance. Overall I think the study could have great impact in the field of cancer biology, with deep implications for understanding mechanistically how population dynamics within tumor cells are orchestrated. Particularly, I find very interesting the identification of a set of drug-susceptible cells that are capable of transitioning to a resistance-primed state. Although I believe these results are interesting, the manuscript is a bit poorly organized. More than a few times I found it troubling to identify the conclusions to certain results or I had to piece them together from other multiple parts of the main text or the discussion. Other times, I felt like the authors jumped to conclusions without clearly describing the results in the main text (even if the figure and experimental design schematics are really helpful to make it cohesive). Altogether, this is not so much a problem with the data/conclusions, but rather the organization and presentation of results. Otherwise, my major concern relates to the supposed identification of “intermediate state” cells. Several data suggest to me that this is not truly an intermediate population. First, authors do not find clear evidences of intermediate states by UMAP or trajectory analysis (although this could be done in further depth). Second, the intermediate states appear to be present (at some rate) in both sorted samples, so defining the origin of these supposedly bi-fate clones in a definitive fashion seems difficult. Third, this population is never isolated/enriched and tested independently to validate their behavior. Fourth, it is not clear to me whether these clones that appear in both “susceptible” and “primed” regions are ever found outside these intermediate states, indistinguishable from other fully susceptible cells. The authors do identify enrichment of EMT/TGFbeta signaling pathway in them and then they use activators to confirm that susceptible-cells can be turned into primed cells by simple addition of TGFb. This last series of experiments, together with the state-switching stochastic model is, to me, the best part of the paper. However, I remain unconvinced that the intermediate state is not in fact a relatively long-lived, clonally-imprinted population of “bipotent” cells. In sum, I believe this is a very good study that will have great impact in the field, but some issues need addressing before publication. My point-by-point comments follow below:

We thank the reviewer for their assessment of our work in the context of cancer biology. The reviewer has pointed out some important considerations regarding the manuscript organization and potential for bi-fate clones. To address these comments, we first restructured the manuscript to build out the results in a more logical manner, which involved moving the PI3K inhibitor results from Fig. 1 into 6. We also clarified the results throughout the text to better justify the conclusions.

The potential that the appearance of state-switching is due to bi-fate clones is an interesting alternative hypothesis; however, our previous work on these cells, as well as further analysis of our new data, shows that state-switching is indeed the cause of lineages appearing in both states. We describe this further below in the section where the reviewer outlines this point more thoroughly.

In the description of the first experiment, at the beginning of the main text, lines 95-117 describe in detail a single-cell lineage tracing to be performed (Fig 1A-C). But then, 119-127 talk about the single-cell clustering results obtained (Fig. 1D). And then lines 129-147 introduce experiments and results with PI3K inhibitors (Fig. 1E-G), and they go on and on, without ever mentioning the lineage tracing data. And, as a reader, I get completely lost. I think these experiments might be a bit out of place in this section. I would perhaps first introduce the single cell data results clearly, then the changes with inhibitors, (Fig 1D-G) and then start with describing methodologically the lineage tracing approach in the next figure (Fig 2). Also, I don't understand how these results (PI3Ki data) fit together at this point in the manuscript. They seem a bit out of place, and perhaps better introduced before the data from Figure 5?

We completely agree with the reviewer's suggestion for improving the paper's organization. Per this suggestion, we have made significant changes to the organization of the paper, including moving the

figures and text into the order recommended by the reviewer. We now start the paper by presenting the scMemorySeq technique and experimental design in WM989 melanoma cells. We then focus on our findings from scRNA-seq, analyzing the drug-susceptible and primed gene expression states (Fig. 1) and showing that these states exist in vivo (Fig. 2). Next, we show how we use barcoding data to identify the drivers of state switching (Fig. 3, 4), followed by our scRNA-seq experiment proving we can drive state switching (Fig. 5). With the state-switching paradigm established we conclude the paper by showing that manipulating state-switching has an impact on drug resistance (Fig. 6). We believe this new organization is much more logical and easy to follow.

If NT5E is a better marker to homogeneously enrich resistant cells, it should be validated against NGFR-high and/or AXL-high cells instead of just to a mixed control (in Supp Fig 1F-G).

We agree that to show NT5E is a good marker we must compare it to other markers. In the initial version of the paper, we showed in Supp. Fig. 1G (now Supp. Fig. 2B) that the top 0.2% of NGFR/EGFR high cells lead to similar amounts of resistance to the top 2% of NT5E high cells.

As recommended by the reviewer, we have now performed additional experiments to directly compare the top 2% of NT5E-expressing cells, the top 0.2% of NGFR-expressing cells, the top 0.2% of EGFR-expressing cells, and the top 0.2% of NGFR/EGFR-expressing cells (the gold standard used in previous papers). We sort out a higher percentage of NT5E-high cells (which is less stringent) because NT5E-high cells make up a larger percentage of the population than NGFR and EGFR-high cells. We sorted these samples and then treated each sample with BRAFi for 3 weeks. This data shows that NT5E performs better than sorting for EGFR or NGFR alone and performs similarly well to sorting for NGFR/EGFR high cells while capturing the whole primed cell population instead of just a subset (Supp. Fig. 2A, B (shown below)).

Supplementary Figures 2A and 2B: Validation of NT5E as a primed cell marker and comparison to EGFR and NGFR. Plots showing the number of resistant cells after sorting by different markers and then treating with BRAFi. The left plot compares all single markers and the right plot compares the gold standard NGFR/EGFR-high cells to NT5E-high cells. Each condition has a minimum of 3 replicates and the error bars are the mean absolute deviation. The control condition consists of the bottom 50% of cells for each marker.

The lack of transitioning cells between the susceptible and primed cluster is interesting, but alternative explanations are not sufficiently tested or discussed. First of all UMAP visualization can be misleading, depending on the PCAs selected. Cell cycle score regression was not performed, and this may be forcing the separation of the few “intermediate” cells around the plot, preventing identification of a more clear transition bridge between susceptible-primed groupings. Also, different representations, such as diffusion component analysis or force-directed layouts might be tested. These are sometimes better at representing distances between the different transitioning cells. Do the authors suggest that increasing the number of sampled cells

would identify the trajectory properly? Or is it also possible that the susceptible-to-primed state transition is not gradual or continuous?

We thank the reviewer for suggesting additional analyses for identifying cells between the susceptible and primed clusters. As suggested by the reviewer, we have now extensively tested different dimensionality reduction methods and analysis approaches to visualize these two different populations and added these plots to Supp. Fig. 1E-G, including regressing out the cell cycle, processing each phase of the cell cycle independently, principal component analysis, diffusion component analysis, and force-directed layout.

Upon regressing out the cell cycle effects, we found that the distinction between the drug-susceptible state and the primed state remained, with a slight increase in the grouping of intermediate cells (Supp. Fig. 1E). Because cell cycle regression might not be completely perfect, we also tested plotting cells from each phase of the cell cycle individually and found that drug-susceptible and primed cells continue to cluster separately for each phase (Supp. Fig. 1F). Alternative data visualizations, including the first two principal components, diffusion component analysis, and force-directed layout, showed more cells positioned between the drug-susceptible and primed states (Supp. Fig. 1G (also shown below)).

Moreover, we assessed various PC combinations in the UMAP analysis and determined that incorporating more than six PCs distinctly separated the two populations. While we acknowledge that optimizing parameters could potentially connect these two groups of cells, we emphasize that dimensionality reduction methods can be highly manipulable. As a result, we base our state assignments on high-dimensionality clustering rather than dimensionality reduction outcomes, defining primed cells as those in Louvain clusters expressing numerous known primed state markers.

To acknowledge the above points in the manuscript we have added the following sentence:

“The distinction between these two states is robust and seen in high dimensional space by Louvain clustering and by multiple dimensionality reduction methods (Supp. Fig. 1A, C, E-H).”

Since multiple visualization methods (PCA, diffusion component analysis, and force-directed layout) all showed cells in between the two major clusters, we questioned whether these cells were the same across the different visualization methods. To determine if they were the same, we established arbitrary thresholds in PC1 to capture cells between the two major groups of drug-susceptible and primed cells, which we refer to as "center cells" (see the first panel in the figure below). We then plotted these center cells using other dimensionality reduction methods and found that the cells between the clusters were indeed the same in both diffusion component analysis and force-directed layout. Interestingly, these cells also appeared in the same areas of the UMAP plot as the intermediate state (although the cells are far from an exact match). This suggests that different visualization methods can be used to estimate which cells may be transitioning.

However, this approach necessitates selecting a visualization method believed to be most accurate, which has no definitive answer, and determining appropriate thresholds in 2D space, which also has no definitive answer (as mentioned, the thresholds used here are arbitrary). As a result, we contend that utilizing lineage data to identify intermediate cells provides a more robust approach. This method allows us to focus on cells that have undergone state changes, and although threshold selection is still required, our metric takes into account the expression of over 500 genes deemed critical to the state-switching process, as they are all upregulated in crossing lineages relative to non-crossing ones.

To emphasize the advantage of using lineage information over 2D representations, we have written the following in the text:

“These rare intermediate state cells are confidently identified using lineage information, but are difficult to accurately identify from the scRNAseq alone (Supp. Fig. 1G).”

While it is possible that increasing cell samples would more clearly identify a crossing trajectory using UMAP, it is not necessarily true since crossing cells will always make up a small fraction of the cells. We posit that the transition between the drug-susceptible and primed states, although potentially rapid, is relatively continuous. To gain insight into the continuity of this process, we can refer to the heat map in Fig. 3J showing the gene expression states of the drug-susceptible, intermediate, and primed states. Here we observe that some genes show a gradual increase or decrease in their expression during state switching, rather than abruptly transitioning from no expression to high expression. Importantly, we do find genes that are specific to the intermediate cell state, including *NFATC2* and *MGP*.

Supplementary Figure 1G: scRNAseq data from Figure 1 plotted with different visualization methods. Cells are colored by their state defined in our analyses.

Figure: scRNAseq data from Figure 1 plotted with different visualization methods showing cells in between the primed and drug-susceptible state based on PC1. Light blue designates the cells identified as the center of PC1. These cells have some overlap with the cells identified by the barcodes (above and in Supp. Fig. 1G), but are largely different. This supports the use of lineage barcodes to precisely identify this population of cells.

In line 174, how is a “primed cell” and a “drug-susceptible” cell defined? This is not clearly described in main text, legends or methods. What I got from reading the manuscript is that they are classified by some form of cluster annotation. Previous publications from these authors have shown that these marker-based clusters (AXLhigh, EGFRhigh) are “enriched” for resistant states, but don’t purify them to homogeneity. Wouldn’t it be more stringent to classify them by a real drug-treatment response in a split of the clones? Is anything preventing this?

The reviewer makes an astute point regarding our terminology and definitions for priming. To address this, we have revised the manuscript to clarify that the “drug-susceptible state” and “primed cell state” are defined by our gene expression analyses. To identify cells for classification as drug-susceptible or primed, we determined which Louvain clusters express high levels of primed marker genes. To make this clear we have added the following to the caption of Supp. Fig. 1A:

“Clusters 2, 8, and 9 are labeled as primed cells based on their expression of known primed marker genes and *NT5E*.”

We connect the primed cell state with the priming behavior by showing that NT5E-high cells are much more likely to survive BRAFi/MEKi than the mixed or NT5E-low population. The reviewer accurately notes that sorting for NT5E (or other markers) only enriches for survival but does not guarantee that all of these sorted cells will survive BRAFi/MEKi. Based on these experiments, we have updated the text to emphasize that cells in the primed gene expression state have an increased likelihood, but not certainty, of surviving BRAFi/MEKi treatment. This is stated in the text as follows:

“Thus, we demonstrated that the marker gene *NT5E* captures the entire cluster of transcriptionally similar cells and that these cells are indeed more likely to be resistant to targeted therapy.”

Regarding the reviewer's suggestion to classify cells using the drug response in a split of clones, we acknowledge the merits of doing this, but did not include it in our experimental design as it would have made our interpretation of the state-switching data more challenging. Specifically, if we use a split of our clones for drug treatment, we would have fewer cells from each lineage captured in the scRNA-seq. This would make it more difficult to accurately know which lineages switched states and which did not. Thus, for the goal of capturing state-switching and memory, we did not include a split of clones treated with drug. Rather, we identified specific markers of these states and used sorting and drug treatment to validate the results (Fig. 1E and Supp. Fig. 2A, B).

The state-switch models used throughout are really useful. However, the first introduction to them is a bit confusing (compared with their later implementation in lines 372-395, where they are more clearly explained). As an example, in line 184, right after discussing the differential exp analysis, the reader is obliged to remember what “from this data” means. And it doesn't actually mean the differential expression analysis discussed in lines 180-183 (just before), it actually means the clonal data analysis from lines 171-179, or does it? This is confusing to me.

We thank the reviewer for pointing out this confusion in the text. We have revised this line to clearly state what data we use in the analysis. This line now reads as:

“To derive the rates of proliferation and state switching from the paired scRNAseq and barcoding data, we used a stochastic two-state model (Supp. Methods 1).”

Many terms are used to define the transitioning drug-susceptible cells. Intermediate state, intermediate cells, crossing-lineage. Some unified terms would help in understanding what is being discussed. In line 211, it is said that the “intermediate cells” would have not been identified using standard scRNAseq techniques. Some evidence of this would be helpful to strengthen this statement (for instance, subclustering, PCA and reclustering analysis of the drug-susceptible clusters), especially since 575 genes are differentially expressed between these and non-transitioning drug-susceptible cells.

We agree with the reviewer that unified terminology would enhance the clarity of the paper. We have added text to more clearly explain that crossing lineages are lineages that are known to change states and that these lineages were used to identify a gene list. This gene list was then used to create an “intermediate state score” to determine which cells express the most important genes for state switching. The cells with the highest intermediate state score are called intermediate state cells. We added new text to clearly explain that intermediate state cells are defined as cells that score highly with the intermediate state score. This is stated in the text as follows:

“We set a threshold on this score and classified the high-scoring cells as “intermediate state cells” (Fig. 3F).”

As suggested by the reviewer, we performed additional analyses to strengthen the statement that “intermediate state cells” would not have been identified with standard scRNA-seq analysis techniques. First, we demonstrate that employing multiple visualization techniques does not reveal a discernible method for delineating intermediate cells using thresholds in 2D space (Supp. Fig. 1F, G). Furthermore, we reanalyzed drug-susceptible cells exclusively and found that intermediate cells neither cluster uniformly in 2D UMAP space nor occupy the same Louvain cluster (see below). This underscores the utility of leveraging lineage information to detect cells in the intermediate state.

Figure: Analysis of only the drug-susceptible cells from the single cell data in figure 1F. The left plot shows cells in UMAP space labeled by their Louvain cluster. The right plot shows the same UMAP plot as the left but labels the intermediate cells in purple.

Determining the initial cell state of the labeled clones by sort-enrichment is probably insufficient and single-cell RNAseq + clone splitting (as in Figure 5) would have been a better way of defining the initial states of cells and their fate-switching. Instead, two different populations were sorted and separately cultured and profiled some time after barcode labeling. But what were these populations really initially (before so many doublings)? If one population was primed-enriched and the other one contains a mix of both susceptible and primed-resistant states, then the origin of the bi-state clones could be misinterpreted. Also, the label on Fig 1C seems misleading, because it suggests that drug-susceptible cells were sort-enriched, and not a “mixed population” as stated in both the figure legend and text. The switching from primed-to-susceptible is clear, but the switch from susceptible-to-primed is not as clear to me if the sorted cells were not relatively pure at starting point. If this is all an issue of AXL/NGFR/EGFR expression heterogeneity, then perhaps NT5E could be used to purify the susceptible cells much better?

Justification for sorting: Overall, sorting has limitations, but given the effect sizes in our data, we can rule out the possibility that the observed state-switching lineages are derived from contamination.

The reviewer is specifically **concerned that drug-susceptible cells do not switch into the primed state**. If this is true, the only explanation for lineages that contain cells in both states would be that they arise from primed cells that are in the mixed sample. Based on our scRNA-seq, the primed cell state makes up 8% of the mixed population. If primed cells were the source of all lineages that contain both states, the maximum number of crossing lineages that we could see would be 8%. However, our data shows 21% of lineages from the mixed sample consist of cells in both states. Thus, some portion of these lineages must have started from a cell in drug-susceptible state, thereby **demonstrating that drug-susceptible cells can switch into the primed state**.

Furthermore, the mixed sample also contains 7% of lineages that are entirely primed. This number is consistent with 8% of lineages starting with a primed cell and many of them remaining entirely primed.

Therefore, the 21% of lineages containing both states described above must predominantly originate from cells in the drug-susceptible state.

The reviewer is also concerned about our choice to derive a mixed sample. We did not sort this sample because the expected percentage of primed cells is relatively low, particularly compared to the expected purity on the sorter. Given the logic above, we know that drug-susceptible cells must be generating crossing lineages.

We also have experimental validation of the drug-susceptible to primed state switching in our prior work using NGFR-reporters that directly show switching (see further discussion of this in the next comment regarding the bi-potent clone hypothesis) (Shaffer et al. 2020).

While it might be possible to provide a more pure isolation of the drug-susceptible cells, we anticipate that this would not change the claims or interpretation of the results. Specifically, it would not change our claim that cells can switch from the drug-susceptible to primed state or the identification of the intermediate cell state using lineages.

The reviewer also brings up the following comment (shown above as well):

If one population was primed-enriched and the other one contains a mix of both susceptible and primed-resistant states, then the origin of the bi-state clones could be misinterpreted.

As described above, the state-switching phenomenon cannot be explained by impurities given that the mixed lineages are at a higher frequency than any possible impurities. Thus, it is not possible that the state-switching phenomenon comes from impurities. The reviewer is also concerned that the intermediate cell state represents a bi-state clone. This intermediate cell state is present in 2% of cells across the entire dataset. Thus, if only 2% of cells could generate both states, we should see only 2% of lineages with mixed states. However, our data shows 18% of lineages contain mixed states (29% of these in the prime-enriched sample and 71% in the mixed sample), **thus mixed lineages occur at a significantly higher frequency than the intermediate cell state**, thereby ruling out the possibility that these intermediate cells are the source of the mixed lineages.

The reviewer is correct about the label in Fig. 1C and this has been fixed in the manuscript.

More worryingly, EGFR, and especially AXL, expression seem to be slightly enriched partially in the bottom right of the susceptible clusters, very close to the location of some of the intermediate/switching states. In fact, in Supp Figure 1B, it seems that those intermediate states are even enriched in the EGFR/NGFR^{high} sample, though maybe a bit hidden by the layering. It would be great to visualize the colors with some transparency, or generate separate plots/quantifications of cluster enrichment for each sample to improve visualization. This all suggests that perhaps “intermediate state” are actually deterministic, heritable, clonally-imprinted bi-potent states, closer to the drug-susceptible region, that are deterministically giving rise to both susceptible and primed states over time. This could be especially concerning if this population is present in both of the sorted samples.

The reviewer accurately notes that EGFR and AXL are slightly upregulated in the intermediate state. We surmise that this occurs because these transitioning cells have not yet fully gained EGFR/AXL expression when switching to the primed state or have not entirely lost expression of EGFR/AXL when switching to the drug-susceptible state. However, the enrichment of sorted primed cells in this region is likely due to the fact that during the expansion of lineages for ~4 doublings we are seeing cells in the early stages of cells switching from the primed state to the drug-susceptible state, and not due to

contamination in the sort. We do not believe it is due to the sort isolating intermediate cells accidentally as we sorted the top 0.2% of NGFR/EGFR expressing cells in the experiment and if we look for the top 0.2% of *NGFR EGFR* expressing cells in our scRNA-seq mixed cell population we see that none of them are in the intermediate state. This suggests that we are not accidentally enriching for many intermediate cells during our sort, and rules them out as the origin of crossing lineages.

The reviewer has pointed out that EGFR and AXL expression is observed at a low to intermediate level in the bottom of the cluster containing drug-susceptible cells. They propose a **hypothesis that these cells represent a bipotent stem cell that is able to generate both the drug-susceptible and primed state**. From both our experimental data in this paper and prior work, we can draw from multiple lines of evidence to rule out this possible model:

1. **The number of intermediate cells is too low to explain state switching.** The strongest line of evidence in our data is the frequency of these intermediate cells. In our scRNA-seq data, they represent about 2% of cells at most. The reviewer is proposing that these cells were in both samples used for our experiments. However, even if this population was in both samples, we capture a much larger percentage, 18% (20% if only looking at the mixed sample), of lineages that switch states overall. Thus, the frequency of the intermediate state is far too low to explain the number of lineages that have switched states.
2. **Clonal dilutions generate both states in all clones.** In our previous work (Shaffer et al. 2017, 2020), we isolated single cells and allowed them to expand to generate clonally derived lines. In these clonally derived lines, we find that all clones contain cells in both the primed and drug-susceptible cell states. If there was a bi-potent progenitor cell, then only the clones derived from this bi-potent progenitor would contain both cell states (and these would be at a lower frequency in the data (2%) if based on the intermediate state).
3. **Direct observation of state switching by time-lapse microscopy.** In our previous work (Shaffer et al. 2020), we tagged the endogenous locus of the *NGFR* gene to generate a fluorescent reporter of the primed cell state. We then performed time-lapse imaging of these cells. We observed that cells fluctuate from a state where they do not express NGFR into a state with high levels of NGFR expression and then back into a state where they do not express NGFR. If the intermediate cells were the source of primed cells, we would expect to see cells with a medium level of NGFR expression that generate cells without NGFR expression or with high NGFR, not the on-off phenotype we observe in time-lapse imaging.

To emphasize this prior work and that state switching has been previously shown, we added an additional line to the text:

“Based on our previous work, we know that both the drug-susceptible and primed states exist in untreated melanoma cells and that cells can fluctuate between these states (Shaffer et al. 2020).”

We have also implemented the reviewer’s visualization suggestions for Supp. Fig. 1B, and it can also be seen below.

Supplementary Figure 1B: scRNA-seq data with cells labeled by their sample in the experimental design. Cells in blue are the cells in the mixed sample and cells in red are sorted primed cells based on the markers NGFR and EGFR.

Do intermediate cells predictably become drug resistant with a higher rate than non-intermediate drug-susceptible cells? Isolating these cells would be really important to validate these results. Perhaps, if I am correct, then NT5E^{low} AXL/EGFR⁺ cells would be a reasonable suggestion to enrich them, since it appears that AXL/EGFR are high at least in some regions of the intermediate state, and NT5E is solely expressed at high levels in the primed clusters.

From our single-cell RNA sequencing data, we can see that 41% of cells in the intermediate state are in lineages that switched state (relative to 29% and 17% for drug-susceptible and primed lineages respectively) suggesting that these cells are indeed much more likely to switch states. We agree with the reviewer that isolating the intermediate cells would be ideal. We spent a significant amount of time and resources trying to isolate these cells using multiple markers (including RAMP1, NTM, THBS1, ANXA1, ALCAM, and ITGB5). However, we struggled to validate antibodies for any of these markers because either the antibody was not specific or the antibody didn't really stain a significant population of cells.

Here, the reviewer proposes a combination of markers consisting of NT5E and AXL or EGFR. Trying to use a combination of cell surface markers is a good idea, however, it is not possible since the intermediate cells are not NT5E^{low}, and are not high for other markers such as AXL, NGFR, and EGFR. Scatter plots of the expression of these markers in cells (based on our scRNA-seq data) show that there is no clear gating strategy to isolate intermediate cells based on these markers.

Figure: Scatter plots of the expression levels of the specified genes to emulate where drug-susceptible, intermediate, and primed cells would appear on a plot during sorting. Intermediate cells are labeled by purple dots and are not a clearly identifiable population using these markers.

Is there a continuous trajectory of gene expression state changes throughout intermediate cells that leads to primed cells? Or do the authors support the idea that this is rather a stochastic gene expression state connected to a higher probability of transition to the resistance-primed state?

The transitions between states are continuous for many of the marker genes. This continuity can be seen if we look at the expression levels of genes across the drug-susceptible, intermediate, and primed

states (see plot below). Furthermore, the heat map in Fig. 3J provides insight into the continuous nature of these gene expression states. We also observe genes uniquely upregulated in the intermediate cell states, such as *NFATC2* and *MGP*.

Figure: Box plot of the specified genes showing their expression level in each cell state. This plot shows that most genes have an intermediate level of gene expression in the intermediate state indicating a continuous transition between the drug-susceptible and primed states, passing through the intermediate state. Of note, *MGP*, *NFATC2*, and *CEBPD* show higher expression in the intermediate cell state.

The drug-induced fate-switching experiment in Figure 5 is really cool. A second untreated sample would have been useful to quantify the baseline rate of lineage conversion of those clones. Maybe the TGFBRi-treated sample serves that purpose to some extent, since there seems to be little to no effect with this compound.

We agree with the reviewer that incorporating a second untreated sample could have been a valuable addition. Our experimental design omitted this sample to minimize the number of cell divisions, thereby increasing our confidence that the majority of cells had not switched states during expansion (prior to applying the different drugs). Nonetheless, we agree that TGFBRi can serve this purpose to a certain extent and have added an additional analysis of this data. When considering lineages with two or more cells in each condition, we observe no statistically significant change in the fraction of primed cells when comparing untreated and TGFBRi treated cells. In contrast, TGFBR1 induces a statistically significant increase, and PI3Ki leads to a statistically significant decrease in the fraction of primed cells (Supp. Fig. 6G (below)).

Supplementary Figure 6G: Box plot of the fraction of primed cells in lineages in each condition. By including all the lineages we show a statistically significant increase in the proportion of primed cells with TGFB1 treatment and a statistically significant decrease in the proportion of primed cells with PI3Ki treatment.

In the clone-splitting experiment, now, for the first time I find the most convincing data regarding the primed-to-susceptible switch. Clearly this is only occurring for a handful of clones here, so it's difficult to assess... however is it possible to compare this data with the bipotent fate data from figure 2 and see the differences? I fear that clonally imprinted AXL/EGFRmid/+ states within the NT5E-low subpopulation of susceptible cells might be at the origin of the bipotent behaviors observed, here identified as "intermediate states". This alternative interpretation needs to be challenged properly. Do switching cells ever produce proper drug-susceptible cells that are indistinguishable from non-switching cells?

We agree with the reviewer that the clone-splitting experiment provides strong confirmation of cells switching between the primed and susceptible states. As the reviewer astutely points out, we only observe a small number of clones that start fully primed, reflecting their low frequency in the population. We emphasize the analysis of lineages that begin entirely in one state because state switching is the only mechanism by which these lineages can change their proportion of primed cells (and not through changes in death or proliferation rates of either state).

We next used a model to gain further confirmation of the state-switching mechanism. To leverage all of the lineage data (not just the lineages that start with cells that are entirely in the drug-susceptible or primed state), we built a model consisting of state switching, growth, and death parameters for a more robust analysis. This model clearly identified state switching as the cause of changes in the proportion of primed cells. Since this data makes us confident that state switching causes the changes in the proportion of primed cells, we can now compare the fraction of primed cells for all the lineages in each condition (instead of limiting the analysis to lineages that started purely in one state as before). By looking at all the lineages this way we can clearly see that there is a significant reduction in the proportion of primed cells when they are treated with PI3Ki (Supp. Fig. 6G (seen above)). In fact, treatment with PI3Ki causes a reduction in the fraction of primed cells for 93% of lineages, indicating that this response is not happening only in a few clones, but rather that it is happening for nearly all of them. We have clarified this in the text by saying the following:

"Of note, 2 lineages did not respond to the PI3K. However, by analyzing all lineages, we find that 93% of lineages reduce their fraction of primed state cells when treated with PI3Ki (Supp. Fig. 6G)."

The reviewer also asks about bipotent behaviors in this comment:

I fear that clonally imprinted AXL/EGFRmid/+ states within the NT5E-low subpopulation of susceptible cells might be at the origin of the bipotent behaviors observed, here identified as “intermediate states”.

See the full discussion above regarding the bi-potent stem cell hypothesis. We can rule out this possibility with multiple lines of evidence: 1. the frequency of switching lineages compared to the intermediate states, 2. reporter cell lines showing bi-directional switching, 3. clonal isolates that are capable of producing each cell state. Together, these experiments rule out the possibility of a low-frequency bi-potent state driving the phenomenon. Furthermore, we show in the plots above that *EGFR/NT5E* combinations do not specifically highlight the intermediate cell states.

The reviewer asks about switching cells and how they compare to the other cell states in this comment: Do switching cells ever produce proper drug-susceptible cells that are indistinguishable from non-switching cells?

In the data, we find that the cells in lineages that switch states are indeed indistinguishable from lineages that maintain their state. The evidence supporting that switching cells are indistinguishable is that cells in state-switching lineages appear in all Louvain clusters, indicating that they adopt all the sub-states within the drug-susceptible and primed states, rather than being restricted to a unique state (see figure below).

To demonstrate that the drug-susceptible cells produced by PI3Ki treatment are not significantly different from naturally occurring drug-susceptible cells, we present a heat map (Fig. 5F) indicating no notable differences between untreated and PI3Ki-treated drug-susceptible cells. To further investigate potential differences, we conducted differential gene expression analysis comparing PI3Ki-treated cells in fully drug-susceptible lineages from the control group to PI3Ki-treated cells in lineages with reduced proportions of primed cells relative to the control group. This analysis aimed to identify any gene expression disparities between cells that became drug-susceptible due to PI3Ki treatment and those that were always drug-susceptible.

Our findings revealed that only two genes, *SERPINE2* and *S100A6*, exhibited slight upregulation in cells transitioning to drug susceptibility. Interestingly, these two genes are highly upregulated in the primed state. As numerous cells that switched states express these genes at similarly low levels as cells that remained drug-susceptible, this minor difference detected is likely attributed to the fact that we captured some cells that had not yet fully downregulated these genes. Overall, this evidence suggests that we can effectively convert primed cells into drug-susceptible cells that are indistinguishable from drug-susceptible cells that did not undergo a transition.

Figure: Bar graph showing the number of cells from different lineage types exist in each Louvain cluster. Switching lineages are those that contain both drug-susceptible and primed cells, drug-susceptible lineages are those that contain only drug-susceptible cells, and primed lineages are those that only contain primed cells.

In Figure 5G, the response of clones to TGFB1 is super heterogeneous. Are there any (transcriptional) differences between the clones that “respond” to TGFB1 and become primed and the clones that do not? Are

they connected to the expression signature of the intermediate state? It would really increase the impact of the manuscript if specific signatures within susceptible cells are present and predict their TGFB1 response.

The reviewer is correct that the response to TGFB1 is not the same across all lineages. This variation could be attributed to some lineages requiring more time to change states. Nonetheless, the response to TGFB1 is relatively consistent in its directionality, as 75% of lineages demonstrate an increase in their proportion of primed cells within the 5 days.

As the reviewer suggested, we performed additional analysis on the lineages that did not respond to TGFB1. By comparing the gene expression of the lineages that did and did not respond to TGFB1, we found no substantial gene expression differences explaining the differences in response. Only one gene was enriched in non-responding compared to responding lineages, which was *SH3BGRL3* (Log2 fold change 0.26).

The modeling of these experiments offers some additional insight into the differences between these lineages. Surprisingly, we found that lineages with a higher proportion of cells already in the primed state were more likely to respond to TGFB1 treatment. Thus, there appears to be something about these lineages that makes them more likely to switch states, but these factors are not captured in our scRNA-seq data.

Reviewer #2: expertise in bioinformatics analysis of transcriptomic data

The authors identify a rare population of melanoma cells which is primed for resistance to treatment from a BRAF and MEK inhibitor combination. The authors characterize the kinetics of the transition between the susceptible and resistant subpopulations using scMemorySeq. They also identify key markers such as NT5E and the EMT-like state to define the primed state or cells transitioning to the primed state. They go on to show that inhibiting PI3K signaling and supplementing TGFB1 ligand can drive cells to switch to susceptible or primed states, respectively. Concerns with this paper relate to the soundness of their conclusion about cellular memory which is critical to the novelty of the work since the presence of such a pathway to cause resistance to BRAF/MEK inhibitor has been well trodden (Smith et al Cancer Cell 2016), and evidence that a subpopulation is responsible has been described (Fallahi-Sichani Mol. Bio.Sys 2017). In the absence of convincing and clear rationale for the concept of cellular memory, it is unclear what else is being added here in the absence of any deep biological or pharmacologic investigation.

We understand the concerns of the reviewer and would like to emphasize the evidence of memory in these cells, the essential role memory plays in our methods, and address the concerns around the novelty of these pathways in the context of melanoma.

First, we would like to highlight the proof of memory in this system, which is shown by multiple lines of evidence. In Fig. 3A, we show that most lineages maintain their gene expression state over time with very few cells changing their cell state. Additionally, in Supp. Fig. 5B-C, we show that the maintenance of cellular memory across cell divisions can be seen at the phenotypic level. In these experiments, we barcoded cells and allowed them to go through 7 doublings. We then split them 4 ways and treated them with BRAFi/MEKi. We found that the same lineages became resistant in each plate, demonstrating that there is memory of the resistance phenotype. This concept is important to the discoveries in this paper, as without cells maintaining states through cell divisions, we would not be able to capture cells undergoing state transitions as illustrated in Supp. Fig. 3E. Furthermore, without

memory, we would not be able to prove state switching in our scMemorySeq experiment, which applies different treatments to the same lineages. This experiment relies on cellular memory to ensure that each lineage, which is split across the different conditions, contains cells that are in the same gene expression and phenotypic state.

Regarding the novelty of the findings, the reviewer references papers that are critical to our understanding of melanoma resistance, but these papers address the changes that occur in cells **after** treatment with targeted therapy. In this manuscript, we focus on using our scMemorySeq approach to study the heterogeneity of cell states and cell state dynamics **before** targeted therapy is added. Further, we show how manipulating these states can lead to better therapeutic outcomes. Some literature has reported the effects of TGFB1 on increasing drug resistance to targeted therapy, but to our knowledge, the ability of PI3Ki to force cells into a more drug-susceptible state has not been reported in the literature. This highlights a novel approach to treating cancers, in which cell states are controlled before targeted therapy to achieve better therapeutic outcomes.

Major Comments

Related to Fig. 1E and Supplementary Fig. 3A-C, the authors indicate that by modulating the number of primed or susceptible cells they can alter the number of eventual resistant cells. However, Supplementary Fig3D shows that different treatment strategies tested can result in differences in cell doublings; these differences may affect endpoint quantification of the number of resistant cells and/or colonies. The authors should show supporting proliferation and apoptotic data of their pretreatment strategies or normalize data to pretreatment controls.

The reviewer makes a very good point that we must account for the effects of differences in proliferation or apoptotic death from the pretreatment. As suggested by the reviewer, we performed additional experiments to quantify the effects of pretreatment on cell number and then normalize our data based on these effects. Specifically, we repeated our pretreatment experiment in Fig. 6A, but added additional wells that we fixed after pretreatment. We then quantified the number of cells in these wells and used that number to normalize the number of resistant colonies, thus allowing us to precisely measure the number of resistant cells relative to the number of cells at the initiation of BRAFi/MEKi. We performed this experiment with 6 technical replicates and 3 biological replicates. With the normalized data, we find that TGFB1 increases resistance and that PI3Ki decreases resistance (by 36%). This result allows us to conclude that although PI3Ki has a modest effect on proliferation, state switching to the drug-susceptible state by PI3Ki is a larger contributor to the reduction in resistance. We have added these normalized counts to the paper (Supp. Fig. 7C) and they can also be seen below. We have also added this information to the main text by saying the following:

In reference to the TGFB1 result:

“Given the potential for the pretreatment time window to affect cell growth, we normalized the resistant data by the number of cells present after pretreatment in each condition. With this normalization, we found that TGFB1 increased the amount of resistance by 2.8 times compared to the control (Supp. Fig. 7B, C).”

In reference to the PI3Ki result:

“We next performed the same normalization as described above for TGFB1. We found that with this normalization, PI3Ki pretreatment decreased the number of resistant cells by 36% (Supp. Fig. 7B, C).”

Supplementary Figure 7C: Box plots showing the normalized number of resistant cells after pretreatment and BRAFi/MEKi. During pretreatment, cells were either untreated, treated with TGFB1, or treated with PI3Ki. After pretreatment, we fixed a sample and counted the number of cells in each condition. The other samples were then treated with BRAFi/MEKi for four weeks. Normalized values are the number of resistant cells divided by the number of initial cells after the pretreatment. P-values were calculated using a Wilcoxon test. TGFB1 increases resistance and PI3Ki decreases resistance upon normalizing for the number of cells present after the pretreatment period.

For example in Figure 1E, the authors claim that the dose used for pretreatment with the PI3K inhibitor did not kill cells nor did it impact cell proliferation; however, in supplementary figure 3D, pretreatment with PI3K is shown to result in decreased cell doublings. The authors should show their analysis for cell proliferation and apoptosis upon PI3K inhibition and show how they derived at their optimal concentration.

The reviewer has raised a valid concern regarding the effect of PI3Ki on cell proliferation. In order to determine the appropriate dose for these experiments, we tested the effects of a range of doses of PI3Ki on proliferation and death using time-lapse microscopy (Supp. Fig. 4A, B (shown below)). To measure cell growth, we counted the number of nuclei over time. To measure cell death, we added a red dye to the media, which would stain the nuclei of cells with compromised membrane integrity. Based on this data, we concluded that the 2µM dose would have minimal effects on cell proliferation and death. We have added data to Supp. Fig. 4A and 4B and updated the text with the following description of the dose:

“Importantly, we selected the dose of 2µM for the PI3Ki as it has minimal effects on cell viability and growth rate and effectively blocks PI3K signaling (Supp. Fig. 4A, B, E).”

It is worth noting that while the 2µM dosage of PI3Ki only has a small effect on proliferation, this seemingly small change does impact the cell counts after 5 days of treatment. These effects can be seen in Supp. Fig. 7A (previously Supp. Fig. 3D), as pointed out by the reviewer. To account for these differences in cell numbers after PI3Ki for 5 days compared to untreated cells, we added new experiments in which we normalize the final number of resistant cells as suggested above (Supp. Fig. 7C).

Supplementary Figure 4A and B: Plots of PI3Ki dose validation. The left plot shows the number of cell doublings over 5 days with different doses of PI3Ki. The right plot shows the doubling of cell death in each condition (doubling is used to normalize for the amount of death at the initial time point).

The authors claim that the primed and susceptible cell states are two discrete states as shown in the two clusters seen in Fig. 1D, however in 5B there does not seem to be a similar discretization in the untreated cells. Moreover, the switch between primed and susceptible seems to be a continuous phenomenon evidenced by the distribution of NT5E expression in the population as shown in Fig. 4B and 5B. Thus it would appear that cells are undulating through a continuum instead of jumping between discrete configurations of those pathways and that some drugs are altering the level of this specific gene signature. Could an alternative model be that NT5E/EMT level in all cells is turned up or down by TGFB or PI3K inhibition, leading to some cells crossing a threshold, making it appear that there is a switch in cell state? What if the histograms in 4B are overlaid, do the drugs inhibit or enhance levels of NT5E in the majority of cells (i.e. shift in median level of NT5E)?

The reviewer proposes that the primed and drug-susceptible gene expression states are actually continuous while we describe them as discrete in the manuscript.

First, we would like to advise the reviewer to exercise caution when interpreting how cells appear in UMAP space. Although these visualizations offer a general sense of the data, there are numerous methods for dimensionality reduction, which may yield different results (Supp. Fig. 1G). For this reason, we employ clustering in high-dimensional space to classify cells as drug-susceptible or primed (specifically, we consider primed cells to be those in Louvain clusters expressing multiple known primed state markers). We refer to the system as discrete because the ultimate phenotype of these cells is binary: they either survive or die upon exposure to targeted therapy. However, at the gene expression level, we do observe a continuous dimension to state-switching (for some genes), which helps us identify the intermediate state. Importantly, while there are some genes that show this continuous pattern of expression, there are other genes that are specific to the intermediate state (including *MGP* and *NFATC2*) and not expressed in either the drug-susceptible or primed.

We agree with the reviewer that TGFB1 and PI3Ki are modulating these cells across a continuum. However, at a point along this continuum, the cells change sufficiently that the phenotype indeed becomes different – as shown in our drug treatment data Fig. 6B. At this point, the cells are functionally different and therefore it is appropriate to define them as changing to either a primed or drug-susceptible state.

As the reviewer suggested, we also performed an additional analysis to measure the shift in NT5E expression using flow cytometry. We found that TGFB1 increases both the mean and median NT5E staining, while PI3Ki decreases both the mean and median. This data further illustrates that TGFB1 and PI3Ki can induce state switching and shift the NT5E expression of the majority of cells (Supp. Fig. 4D).

Supplementary Figure 4D: Density plot of flow cytometry data showing NT5E expression in cells treated with their respective condition. The mean and median are labeled for each condition. The shift in the median indicates that there is a population-level change in NT5E expression with each treatment, not just a large change in a small subpopulation of cells.

In Fig. 2D, MITF is downregulated in sister cells of primed cells compared to lineages without primed cells. Also, in Fig. 2J the primed vs susceptible heatmap shows that MITF is further downregulated in primed cells. It is very well established in the literature that BRAF/MEKi resistant cells in melanoma upregulate MITF (Smith et al, 2016 Cancer Cell, Johannssen Nature 2013 among others). What is the reason for this discrepancy, is there a process where the full resistant cells go from downregulating MITF to then upregulating it?

The reviewer is correct that our data shows that *MITF* is downregulated in cells in the primed gene expression state. It is important to note that the states we describe in our paper occur before the administration of targeted therapy, whereas most previous literature focuses on states following treatment with targeted therapy. We find that the *MITF*-high and *MITF*-low gene expression states exist prior to treatment and can predict which cells will develop resistance upon treatment (Supp. Fig. 2A, B).

In both papers referenced here (Smith et al. 2016; Johannessen et al. 2013), the authors examine *MITF* expression in cells after they have been treated with MAPK inhibitors. Furthermore, although these papers describe a *MITF*-high resistant state, there is also substantial evidence in the literature that resistant cells can be *MITF*-low (Müller et al. 2014; Rambow et al. 2018; Karras et al. 2022). This highlights an interesting feature of melanoma that different gene expression states can be observed in resistant cells.

In the model we employ here, we have evidence that the *MITF*-low cells are the ones that will become resistant to targeted therapy once it is applied. Since we do not perform RNAseq on these cells after drug treatment, we do not have any evidence to support *MITF* eventually becoming upregulated (although this is possible). Similar patterns of *MITF* expression were also observed in our prior work as well, including Shaffer et al Nature 2017 and Emert et al Nature Biotech 2021 (Shaffer et al. 2017; Emert et al. 2021). Interestingly, previously published CRISPR screens using these same cell lines did identify genetic knockouts that could generate resistance with *MITF*-high expression (Torre et al. 2021). Thus, while our primed population is *MITF*-low before treatment, it is clear that resistance to BRAFi/MEKi can occur with MITF-high or *MITF*-low states.

Minor comments:

The NT5E high cutoff for untreated cells used in Fig. 4B should be changed such that the ratio of primed to susceptible cells is the same as shown in the two clusters in both Fig. 1D and Fig 2B for consistency. Ideally the cutoff should be based on the percent of cells that survive BRAFi/MEKi treatment alone.

As suggested by the reviewer, we have added a new analysis in which the NT5E-high cutoff is based on the percentage of primed cells in Fig. 1F. In Supp. Fig. 4C, we now show an 8% cutoff as this is the predicted percentage of primed cells from our scRNA-seq data. Importantly, using this cutoff, our conclusions regarding TGFB1 and PI3Ki remain unchanged (Supp. Fig. 4C (shown below)). For figure 2C, we use a 2% threshold because we have phenotypically validated this threshold. For this validation, we sorted the top 2% of NT5E expressing cells and found that they were more resistant to targeted therapy (Supp. Fig. 2B).

Supplementary Figure 4C: Box plot of flow cytometry data generated with an 8% cutoff for primed cells in the untreated sample. In this plot, a cell is considered to be primed if its NT5E expression is greater than or equal to the top 8% of the untreated cells. This percentage is based on the number of primed cells in the scRNA-seq data shown in Fig. 1D. The other box plots show the percentage of primed cells across different conditions using the same cutoff in NT5E to define a primed cell.

In the case of phenotype switching of barcodes from primed to susceptible and vice versa (Fig. 5G and 5H), the fraction of cells from the respective lineages in the primed state for the untreated sample should be shown for better readability of data.

This is a valuable suggestion from the reviewer and we have added these conditions to the figure (Fig. 5G and 5H).

Figure 5G and 5H: Phenotype switching diagrams and data. On the left (G) is a schematic of lineage-based analysis to test for state switching into the primed state. Box plots show the fraction of cells in each lineage that are in the primed state across all the conditions. The lineages shown on this plot are exclusively those that were completely drug-susceptible in the untreated sample. On the right (H) is a schematic of lineage-based analysis to test for state switching into the drug-susceptible state. Box plots show the fraction of cells in each lineage that are in the primed state across all the conditions.

In Fig. 3A, the authors should quantify the expression of SOX10 and NT5E in the region of interest compared to a control region. Fig. 3B and Fig 3D require controls (normal tissue or untransformed samples) for primed and susceptible states in order to show that the heterogeneity captures something outside the normal range of expression.

The reviewer raises an important point regarding the value of comparing expression differences to a control. In the case of HCR FISH in tissue, since these are melanoma cells in a mouse PDX model, we do not have paired human WT melanocytes within this model. However, our data does effectively demonstrate much higher levels of *NT5E* and significantly lower expression of *SOX10* in these rare cells compared to other cells in the tumor. Here, the other cells within the tumor serve as the control for comparison.

While we do not have wild-type human melanocytes in our mouse model. We leveraged scRNA-seq of wild-type human melanocytes (Belote et al. 2021) to further address this point. We used the primed cell gene expression state and scored both melanoma biopsies and WT melanocytes. We found that the range of primed cell scores is significantly larger in melanoma biopsies compared to WT melanocyte biopsies, thus supporting that this heterogeneity would not be observed in melanocytes (Supp. Fig. 3B (shown below)). This suggests that the high primed cell signature in melanoma represents meaningful biological differences that do not occur in normal melanocytes.

Supplementary Figure 3B: Box plot showing the range of primed gene set enrichment scores per patient in melanocytes versus melanoma. This data shows that the differences in primed gene set enrichment score seen in melanoma represent a meaningful difference in cell state that is not present in wild-type melanocytes.

In Fig. 5G, a better visualization of the concept shown would be to take a representative number of lineages (say 20 lineages) and connect them via a line across the different treatments. This would show how each specific lineage changes in the fraction of primed cells by treatment.

We thank the reviewer for this suggestion and generated an additional supplementary figure to show the type of visualization requested (Supp. Fig. 6H (shown below)).

Supplementary Figure 6H: Paired dot plots showing example lineages change their proportion of primed cells based on treatment. Each point and line color corresponds to the same lineage. This plot shows that the large majority of 20 random lineages plotted increase their proportion of primed cells when treated with TGFB1 and decrease their proportion of primed cells when treated with PI3Ki.

The authors should show that they can recapitulate the two clusters observed in Fig. 1D UMAP from the untreated cells in Fig. 5B,C,D to better define the primed and susceptible populations. Can you take the annotations from this newly created UMAP and map them onto the UMAP shown in Fig. 5B,C,D to see if it aligns with your definition of primed and susceptible populations shown in that figure?

The reviewer makes a good point that we should validate that the untreated primed cells in Fig. 5 are similar to those in Fig. 1. We do note that although the UMAP representation can be helpful to understand the data, the level of separation between clusters of cells in UMAP is not easily interpretable. Thus, it is more informative to look at high-dimensional Louvain clustering to identify

states. To that end, we now describe how the identity of primed cells was determined based on identifying Louvain clusters with high primed gene set signature score (Supp. Fig. 6C (shown below)). We say this in the text in the following manner:

“To classify cells as primed or drug-susceptible, we selected Louvain clusters 4 and 10, as these clusters contained the majority of cells expressing the known primed state marker genes (Fig. 5D, Supp. Fig. 6B, C).”

Supplementary Figure 6C: Box plot of the primed gene set signature score of cells in each Louvain cluster. Clusters 4 and 10 were classified as primed clusters since they have much higher primed gene set signature scores than the other clusters.

Furthermore, to show that these states are the same across sequencing runs, we plotted all untreated WM989 samples into one UMAP plot (from the experiments in Fig. 1, Fig. 5, and Supp. Fig. 1H). We maintained the annotations as primed or drug-susceptible as determined in each plot individually and show that the primed cells still all group together (figure below).

Figure: UMAP of all untreated WM989 cells sequenced in the paper. Cells are colored using the primed and drug-susceptible labels determined in each individual experiment.

Given that primed and susceptible cells have different growth rates, it may be reasonable to assume that there might be differences in the cell cycle which could explain the clusters in Fig 1D. The data should have cell cycle markers regressed out. Supplementary data Fig. 2B shows a high difference in % of cells in S phase between the two clusters. If the two clusters no longer remain after regressing out those markers, then there is a significant flaw in using this method to call cells as primed or drug-susceptible.

The reviewer has a valid concern that the cell cycle may be driving the different states. We have added a supplementary figure regressing out the cell cycle and plotting cells from each of the cell cycle phases individually (Supp. Fig. 1E, F (shown below)). Each of these plots still captures the two states, demonstrating that the states exist across cells in different phases of the cell cycle.

Supplementary Figure 1 E and F: The separation between the primed and drug-susceptible state cells is not dependent on the cell. The first plot shows a UMAP projection after cell cycle regression. Cells are colored by their phase in the cell cycle. The three plots on the right are UMAP plots with cells from each phase of the cell cycle plotted individually. In these plots, cells are colored by their gene expression state.

Reviewer #3: expertise in transcriptional regulation

Summary:

In this interesting study, Harmange and colleagues developed a single-cell RNA-sequencing based approach to measure cell state memory in mammalian cells. This is of clinical relevance as it has been previously shown that melanoma cells can switch between therapy resistance (primed) and drug-naïve cell states. To measure cellular memory the authors aimed to track transcriptional states across cell lineages. Towards this, Harmange et al applied a barcoding library to melanoma cells and subsequently profiled the transcriptome of those barcoded cells at a single-cell level after a certain number of cell doublings. Through lineage tracing with cell barcoding the authors found that most cells maintained their cell state memory, which was in contrast to their previous conclusions from bulk RNA-sequencing studies that indicated dynamic fluctuation between therapy resistant and drug-naïve gene expression states in the same studied cell line. However, in their new manuscript using single-cell analysis the authors identified a pool of cells that lost their gene expression memory and they demonstrated that this lineage switching is dependent on PI3K and can be induced by TGFB1. They also confirmed that similar gene expression signatures exist in mouse models and patient samples. Finally, the authors demonstrated that the transcriptional memory remains stable over time and that treatment with TGFB1 and PI3Ki induces primed and drug-susceptible states, respectively. Although only a small fraction of cells switches states upon those treatments this is still sufficient to alter the outcome of treatment regimens across a cell population.

Overall, the experimental approach (scMemorySeq) and conclusions of this study are novel, accurate, clinically relevant and of great interest to a broad readership. The experiments are well controlled, data are presented in a clear and visually attractive manner and methods are described in sufficient details. Overall, this manuscript is suitable for publication after addressing the minor comments listed below.

Minor comments:

- Line 50-53: It would be helpful if the authors could summarize their previous work in a bit more detail. Maybe they could add 1-2 sentences briefly describing the technology and cancer type used along with major finding from the cited studies.

Thank you for the suggestion. We have added some more information about the findings of previous work to contextualize our findings at the start of the results section. We added the following:

“Based on our previous work, we know that both the drug-susceptible and primed states exist in untreated melanoma cells and that cells can fluctuate between these states (Shaffer et al. 2020). When

targeted therapy is applied, cells in the primed state have a higher likelihood of resistance, whereas cells in the drug-susceptible state succumb to the treatment (Shaffer et al. 2017).”

- Line 242: delete “population”

Corrected, thank you.

- Line 335: Shouldn't this read “SOX-high” instead of “SOX-low”?

The reviewer is correct, we have fixed this in the text.

- The authors apply a PI3K inhibitor and show that this decreases drug resistance in melanoma cells. PI3K inhibition should ideally be confirmed by assessing phosphorylation state of kinase targets by western blotting.

The reviewer makes a valid point that we should test how the PI3Ki works in our melanoma cells. We performed additional experiments in which we used a western blot to show that the PI3Ki can block activation of the PI3K pathway, even in the presence of insulin growth factor (IGF) stimulation. Moreover, we show that the PI3Ki eliminates PI3K signaling in the primed cell state (Supp. Fig. 4E (below)).

Supplementary Figure 4E: Western blot showing phospho AKT. In the labels, mix is an unsorted sample of WM989 cells. Primed is the top 2% of NT5E-expressing cells. This data shows that the PI3Ki used at 2 μ M effectively blocks PI3K signaling.

- Also, can the authors comment on whether addition of TGFB1 induce any markers of EMT as observed in intermediate cell states (using scRNA-seq data from Fig. 5)?

Yes, TGFB1 induces markers of EMT. To make this clear we have mentioned in the text and added a supplementary figure showing an increased amount of EMT gene expressed in TGFB1 treated cells (Supp. Fig. 6D (shown below)). We have added the following to the text:

“Importantly, the primed cell state induced by treatment with TGFB1 was transcriptionally very similar to untreated cells in the primed state, and induced EMT and TGF- β signaling genes (Fig. 5F, Supp. Fig. 6D, E).”

Supplementary Figure 6D: Box plot of the EMT gene set signature score of cells in each treatment condition. The EMT signature score is considerably higher in the TGFB1-treated cells, indicating that TGFB1 increases the expression of EMT genes.

- Given the prominent association with primed cell state, the authors should consider displaying the NT5E marker gene in the heat maps displayed in Fig. 5F. Do the NT5E transcript levels correlate with the protein level shown in Fig. 4C?

NT5E is a row in the heatmap in Fig. 5F, but it was previously difficult to discern. We have now improved the clarity of the labels for better visualization. Overall, we find that sorting cells based on their NT5E protein levels indeed enriches for primed cells that become resistant when therapy is applied. We compare this to previous markers including EGFR and NGFR that are also expressed in primed cells. We find that all three of these markers are present in the primed state by scRNA-seq and enrich for the cells that will become when drug is applied (Fig. 1D, Supp. Fig. 2A). Thus, these results indicate that both protein and RNA are high in primed cells for the primed cell markers NT5E, NGFR, and EGFR.

Reviewer #5 (Remarks to the Author): expertise in melanoma drug resistance

In their manuscript, Harmange et al present a novel method for in vitro tracking of phenotypic switching using barcoding and scRNAseq. Using this method, they explore the kinetics and dynamics of cellular switching between a drug-sensitive state and a state primed for drug resistance in melanoma. Authors then examine how pre-treating cells with inhibitors of TGFBR and PI3K based on the transcriptional characteristics of resistance “primed” states affect these transitions. These data provide insights into the molecular pathways which may regulate the transitions between cell states. Data gathered in these studies was then utilized to parameterize a mathematical model of phenotypic switching between the drug-susceptible and drug-primed state. Although many of the transcriptional characteristics of the drug-sensitive and drug-resistant states have been defined previously, the kinetics and dynamics of cell transitions through these states have never been explored and are of high interest to the melanoma research community. This work also provides a deeper functional dimension to some of the pathways previously implicated in BRAF inhibitor resistance.

Questions/concerns:

1. Is it possible that melanoma tumors have more than 1 drug resistance “primed” and more than 1 drug-sensitive state? There have been a few papers now that have identified 4-6 transcriptional states in melanoma, most recently a paper from the Marine lab in Nature volume 610, pages 190–198 (2022). Is there any indication how the drug sensitive and drug “primed” states defined in this manuscript fit within the 6 cellular state landscape described in the Marine lab study? It would be helpful to understand how the states identified

here fit into the published states. For example, is the “primed” state the same cells that are identified as “mesenchymal-like”? and are the sensitive states a combination of all others? Or are mesenchymal-like the very rare subset of the “primed” state that are then also resistant to PI3Ki pre-treatment coupled with BRAF inhibitor treatment?

The reviewer makes a valuable suggestion to more clearly relate our findings to previously defined states in melanoma. Importantly, most of the states defined in the melanoma resistance literature are derived from cells that have already been exposed to targeted therapy. A key distinction is that our data is from drug-naive cell states that exist prior to treatment but are predictive of treatment response. Bearing this in mind, we compared the expression states found in two papers, (Rambow et al. 2018) and the suggested Karras et al. 2022 paper (Karras et al. 2022). By comparing these with our own data, we can see that the primed cell state is a mix of the invasive, neural crest-like, stress-like, and mesenchymal-like states. From this analysis, we can see that primed cells express genes from multiple of these categories, but these categories do not clearly identify subsets of our primed state. Rather the primed state is some mixture, which may have its own further subclassification that does not follow those in these papers. The text now clearly states that there are similarities between states found in this paper and previous work by saying the following:

“We also found that primed cells have similarities to previously published gene expression states associated with drug resistance in melanoma (Supp. Fig. 2C) (Rambow et al. 2018; Karras et al. 2022; Müller et al. 2014; Zuo et al. 2018; Sun et al. 2014; Capparelli et al. 2022; Smith et al. 2016; Ji et al. 2015; Girotti et al. 2013).”

We have also added a figure showing the gene sets of the states found in these papers on our data in Supp. Fig. 2C which we show below:

Supplementary Figure 2C: UMAP plots showing the gene set signature score of different states found in the literature in WM989 cells. These plots demonstrate that the primed state represents a combination of states previously described in the literature, but that it does not perfectly align with any one of these.

2. Is pre-treatment with PI3K more, less, or similarly as effective compared to simultaneous combination treatment? This is an essential question that needs to be answered and would have a large impact on how dynamic treatment strategies are developed preclinically and clinically.

This is a good suggestion by the reviewer to compare cotreatment with PI3Ki compared to pretreatment. We performed cotreatment experiments in which we treated cells with BRAFi/MEKi/PI3Ki and compared them with cells pretreated for 5 days with PI3Ki and then treated with BRAFi/MEKi. We found that treating with all three inhibitors at the same time eliminated almost all cells and performed better than pretreatment. However, this is a highly toxic combination that has not been possible in patients due to side effects. Generally, blocking both the MAPK pathway and PI3K pathway at the same time is highly toxic, which has required other studies to use low doses of these drugs limiting the effectiveness of this otherwise promising treatment (McNeill et al. 2017; Bardia et al. 2020; Shapiro et al. 2020). By using the pretreatment method, we may be able to get some of the effectiveness of the combination treatment by using higher doses of drugs while minimizing toxicities. We have now included this data in the text by saying the following:

Furthermore, treating cells with PI3Ki and targeted therapy at the same time was even more effective than pretreatment, nearly eliminating all resistance (Supp. Fig. 7D, E). Although blocking the PI3K pathway at the same time as the MAPK pathway is effective at killing melanoma cells, it is toxic to patients which has forced clinical trials to use low doses of these drugs, thus limiting the effectiveness of this otherwise promising combination (McNeill et al. 2017; Bardia et al. 2020; Shapiro et al. 2020)."

We have also added this data in Supp. Fig. 7D and E (shown below).

Supplementary Figure 7D and E: Cotreating with BRAFi, MEKi, and PI3Ki dramatically decreases resistance. On the left are representative wells showing the nuclei (shown as black dots) after the specified treatment. On the right is the quantification of the data across 3 biological replicates, each with 6 technical replicates.

3. Importantly, the authors show that pre-treating cells with PI3K significantly reduces but does not completely abolish the outgrowth of resistant cells. The authors should analyze the transcriptional profiles of rare cells in the lineage which did not respond to PI3Ki. It is essential to know more about these states, how they are different from BRAFi-sensitive and "primed" states, and potential insights into how they can be targeted.

We agree with the reviewer that a clear gene expression distinction between cells in lineages that respond to PI3Ki treatment and those that do not would be insightful. To uncover such differences, we performed a differential gene expression analysis on cells in lineages that responded to PI3Ki treatment versus those that did not. Unexpectedly, we found minimal significant differences. Specifically, we only

find one gene *LMO2* which is slightly upregulated in non-responding lineages and which has no clear role in state switching. We have added these new analyses to the paper in Supp. Table 3, and added the following to the text:

“Moreover, there were no systematic gene expression differences that explained the differences in responsiveness to PI3Ki (Supp. Table 3).”

4. If the drug-susceptible cells in lineages with a high proportion of cells already in the primed state are more easily switched into the primed state, is it possible to predict clinical response to BRAFi based on the content of the primed state in a tumor?

We agree with the reviewer that determining the proportion of primed cells in a tumor before treatment with BRAFi would be a useful way to evaluate the clinical relevance of primed cells. This analysis was previously done by Konieczkowski et al. where they showed that patients that had MITF-/AXL+ (primed cells) in their drug-naive biopsies had a shorter progression-free survival on BRAFi/MEKi than patients with MITF+/AXL- (drug-susceptible) cells (Konieczkowski et al. 2014). This finding highlights the potential clinical benefit of reverting the primed state before the addition of BRAFi/MEKi clinically. We have added this information to the text by saying the following:

“Furthermore, a previous analysis of data from the cancer genome atlas showed that the presence of MITF-low/AXL-high cells (as also seen in our primed cell state) in drug-naive patient tumors was predictive of a shorter progression-free survival rate (Konieczkowski et al. 2014). Together, these data demonstrate the generalizability of the primed cell state and suggest that it might be predictive of response to BRAFi/MEKi.”

5. The authors are tackling a phenomenon that has been studied intensely for the last decade. It is important to be more thorough about setting the context for this study and make sure cite additional papers related to:

- a) Reversal to drug-sensitive state and leveraging of state dynamics in therapy scheduling
- b) AXL/MITF/SOX10/FN1 transcriptional states related to BRAFi resistance
- c) Role of TGFbeta in BRAFi response
- d) Role of PI3K inhibitors in reversing BRAFi resistance

Although authors do cite some work in some of these areas, it could be more thorough.

We thank the reviewer for this guidance in framing our study and have added a more thorough discussion of the context for this work in both the introduction and discussion. For each of these categories outlined, we have added additional papers to strengthen the context of our paper.

Minor questions/concerns:

1. Why did almost half the cells profiled not have barcodes (related to results for Figure 1)? Are barcodes lost over time or were barcoded and non-barcoded cells analyzed?

There are multiple reasons why cells may not have a barcode, and the percentage of recovered barcodes is consistent with what others have been able to do (Raj et al. 2018; Bowling et al. 2020). The most likely place to lose a barcode is during the preparation of the 10X libraries. Barcodes are captured in the same way other transcripts in the cell are captured, which is by binding the poly-A tail on the mRNA transcript, but this process only samples a small percentage of the transcripts of the cells. Thus, it is possible that in the sampling of the transcriptome, a barcode transcript never gets captured. Also, even if a lineage barcode is captured in this first step, it may not get amplified as amplification of a transcript requires successful template switching for primers to bind, which is not an efficient process.

Finally, to avoid false positives, we set very conservative cutoffs throughout our analysis pipeline on what we confidently call a lineage barcode. While this ensures that we are confident in the lineages that we do identify, it is possible that we are also losing barcode data from other cells.

2. There is a typo on line 24

Thank you, we have corrected this typo in the text.

References

- Bardia, Aditya, Mrinal Gounder, Jordi Rodon, Filip Janku, Martijn P. Lolkema, Joe J. Stephenson, Philippe L. Bedard, et al. 2020. "Phase Ib Study of Combination Therapy with MEK Inhibitor Binimetinib and Phosphatidylinositol 3-Kinase Inhibitor Buparlisib in Patients with Advanced Solid Tumors with RAS/RAF Alterations." *The Oncologist* 25 (1): e160–69.
- Belote, Rachel L., Daniel Le, Ashley Maynard, Ursula E. Lang, Adriane Sinclair, Brian K. Lohman, Vicente Planells-Palop, et al. 2021. "Human Melanocyte Development and Melanoma Dedifferentiation at Single-Cell Resolution." *Nature Cell Biology* 23 (9): 1035–47.
- Bowling, Sarah, Duluxan Sritharan, Fernando G. Osorio, Maximilian Nguyen, Priscilla Cheung, Alejo Rodriguez-Fraticelli, Sachin Patel, et al. 2020. "An Engineered CRISPR-Cas9 Mouse Line for Simultaneous Readout of Lineage Histories and Gene Expression Profiles in Single Cells." *Cell* 181 (6): 1410–22.e27.
- Capparelli, Claudia, Timothy J. Purwin, Mckenna Glasheen, Signe Caksa, Manoela Tiago, Nicole Wilski, Danielle Pomante, et al. 2022. "Targeting SOX10-Deficient Cells to Reduce the Dormant-Invasive Phenotype State in Melanoma." *Nature Communications* 13 (1): 1381.
- Emert, Benjamin L., Christopher J. Cote, Eduardo A. Torre, Ian P. Dardani, Connie L. Jiang, Naveen Jain, Sydney M. Shaffer, and Arjun Raj. 2021. "Variability within Rare Cell States Enables Multiple Paths toward Drug Resistance." *Nature Biotechnology* 39 (7): 865–76.
- Girotti, Maria R., Malin Pedersen, Berta Sanchez-Laorden, Amaya Viros, Samra Turajlic, Dan Niculescu-Duvaz, Alfonso Zambon, et al. 2013. "Inhibiting EGF Receptor or SRC Family Kinase Signaling Overcomes BRAF Inhibitor Resistance in Melanoma." *Cancer Discovery* 3 (2): 158–67.
- Ji, Zhenyu, Yiyin Erin Chen, Raj Kumar, Michael Taylor, Ching-Ni Jenny Njauw, Benchun Miao, Dennie T. Frederick, et al. 2015. "MITF Modulates Therapeutic Resistance through EGFR Signaling." *The Journal of Investigative Dermatology* 135 (7): 1863–72.
- Johannessen, Cory M., Laura A. Johnson, Federica Piccioni, Aisha Townes, Dennie T. Frederick, Melanie K. Donahue, Rajiv Narayan, et al. 2013. "A Melanocyte Lineage Program Confers Resistance to MAP Kinase Pathway Inhibition." *Nature* 504 (7478): 138–42.
- Karras, Panagiotis, Ignacio Bordeu, Joanna Pozniak, Ada Nowosad, Cecilia Pazzi, Nina Van Raemdonck, Ewout Landeloos, et al. 2022. "A Cellular Hierarchy in Melanoma Uncouples Growth and Metastasis." *Nature*. <https://doi.org/10.1038/s41586-022-05242-7>.
- Konieczkowski, David J., Cory M. Johannessen, Omar Abudayyeh, Jong Wook Kim, Zachary A. Cooper, Adriano Piris, Dennie T. Frederick, et al. 2014. "A Melanoma Cell State Distinction Influences Sensitivity to MAPK Pathway Inhibitors." *Cancer Discovery* 4 (7): 816–27.
- McNeill, Robert S., Demitra A. Canoutas, Timothy J. Stuhlmiller, Harshil D. Dhruv, David M. Irvin, Ryan E. Bash, Steven P. Angus, et al. 2017. "Combination Therapy with Potent PI3K and MAPK Inhibitors Overcomes Adaptive Kinome Resistance to Single Agents in Preclinical Models of Glioblastoma." *Neuro-Oncology* 19 (11): 1469–80.
- Müller, Judith, Oscar Krijgsman, Jennifer Tsoi, Lidia Robert, Willy Hugo, Chunying Song, Xiangju Kong, et al. 2014. "Low MITF/AXL Ratio Predicts Early Resistance to Multiple Targeted Drugs in Melanoma." *Nature Communications* 5 (December): 5712.
- Raj, Bushra, Daniel E. Wagner, Aaron McKenna, Shristi Pandey, Allon M. Klein, Jay Shendure, James A. Gagnon, and Alexander F. Schier. 2018. "Simultaneous Single-Cell Profiling of Lineages and Cell Types in the Vertebrate Brain." *Nature Biotechnology* 36 (5): 442–50.
- Rambow, Florian, Aljosja Rogiers, Oskar Marin-Bejar, Sara Aibar, Julia Femel, Michael Dewaele, Panagiotis Karras, et al. 2018. "Toward Minimal Residual Disease-Directed Therapy in Melanoma." *Cell* 174 (4): 843–55.e19.
- Shaffer, Sydney M., Margaret C. Dunagin, Stefan R. Torborg, Eduardo A. Torre, Benjamin Emert, Clemens Krepler, Marilda Beqiri, et al. 2017. "Rare Cell Variability and Drug-Induced Reprogramming as a Mode of Cancer Drug Resistance." *Nature* 546 (7658): 431–35.
- Shaffer, Sydney M., Benjamin L. Emert, Raúl A. Reyes Hueros, Christopher Cote, Guillaume Harmange, Dylan L. Schaff, Ann E. Sizemore, et al. 2020. "Memory Sequencing Reveals Heritable Single-Cell Gene Expression Programs Associated with Distinct Cellular Behaviors." *Cell*, July. <https://doi.org/10.1016/j.cell.2020.07.003>.
- Shapiro, Geoffrey I., Patricia LoRusso, Eunice Kwak, Susan Pandya, Charles M. Rudin, Carla Kurkjian, James M. Cleary, et al. 2020. "Phase Ib Study of the MEK Inhibitor Cobimetinib (GDC-0973) in Combination with

- the PI3K Inhibitor Pictilisib (GDC-0941) in Patients with Advanced Solid Tumors.” *Investigational New Drugs* 38 (2): 419–32.
- Smith, Michael P., Holly Brunton, Emily J. Rowling, Jennifer Ferguson, Imanol Arozarena, Zsafia Miskolczi, Jessica L. Lee, et al. 2016. “Inhibiting Drivers of Non-Mutational Drug Tolerance Is a Salvage Strategy for Targeted Melanoma Therapy.” *Cancer Cell* 29 (3): 270–84.
- Sun, Chong, Liqin Wang, Sidong Huang, Guus J. J. E. Heynen, Anirudh Prahallad, Caroline Robert, John Haanen, et al. 2014. “Reversible and Adaptive Resistance to BRAF(V600E) Inhibition in Melanoma.” *Nature* 508 (7494): 118–22.
- Torre, Eduardo A., Eri Arai, Sareh Bayatpour, Connie L. Jiang, Lauren E. Beck, Benjamin L. Emert, Sydney M. Shaffer, et al. 2021. “Genetic Screening for Single-Cell Variability Modulators Driving Therapy Resistance.” *Nature Genetics* 53 (1): 76–85.
- Zuo, Qiang, Jing Liu, Liping Huang, Yifei Qin, Teresa Hawley, Claire Seo, Glenn Merlino, and Yanlin Yu. 2018. “AXL/AKT Axis Mediated-Resistance to BRAF Inhibitor Depends on PTEN Status in Melanoma.” *Oncogene* 37 (24): 3275–89.

REVIEWERS' COMMENTS

Reviewer #1 (Remarks to the Author):

The manuscript by Harmange et al. has been significantly improved following the suggestions from multiple reviewers. The new manuscript is a streamlined version that is easier to read and interpret, at least in my opinion. The authors have also responded to most of my criticisms in a successful way. My most important concern, regarding "bipotency", has been successfully addressed. However, there is still the distinct possibility that a subset of clones have a higher heritable likelihood of state-switching, which is being identified (enriched) here as the "intermediate state". How can the authors discard this possibility?

Reviewer #3 (Remarks to the Author):

The paper is improved by the further analysis, particularly the cell cycle normalization. I also agree the novelty of the paper lies in the analysis of heterogeneity before treatment, as opposed to previous papers that have focused on targeting drivers that emerge after treatment. I believe this could be made more clear, potentially in the abstract

Reviewer #4 (Remarks to the Author):

The authors have addressed all my comments and I recommend publication in Nature Communications.

Reviewer #5 (Remarks to the Author):

The authors have sufficiently addressed my comments and concerns in the revised manuscript. Based on this, I recommend this manuscript be accepted for publication.

REVIEWERS' COMMENTS

Reviewer #1 (Remarks to the Author):

The manuscript by Harmange et al. has been significantly improved following the suggestions from multiple reviewers. The new manuscript is a streamlined version that is easier to read and interpret, at least in my opinion. The authors have also responded to most of my criticisms in a successful way. My most important concern, regarding "bipotency", has been successfully addressed. However, there is still the distinct possibility that a subset of clones have a higher heritable likelihood of state-switching, which is being identified (enriched) here as the "intermediate state". How can the authors discard this possibility?

We thank the reviewer for their comments. We agree that the manuscript is streamlined and easier to read. We also agree with the reviewer that it is possible that a subset of the clones have a higher likelihood of state-switching, and that this could be heritable and represented by the intermediate state. Since we can not rule out this possibility, we have added an additional line to the text to communicate this possibility to the reviewer. This line is as follows:

While we observe many lineages that contain cells from both states (3 and 4), it is possible that a subset of cells have a higher propensity for state switching and generate these mixed state lineages.

Reviewer #3 (Remarks to the Author):

The paper is improved by the further analysis, particularly the cell cycle normalization. I also agree the novelty of the paper lies in the analysis of heterogeneity before treatment, as opposed to previous papers that have focused on targeting drivers that emerge after treatment. I believe this could be made more clear, potentially in the abstract

We thank the reviewer for this suggestion and updated the abstract to emphasize this point of novelty. This is the updated line:

Applied to melanoma cells without therapy, we quantify long-lived fluctuations in gene expression that are predictive of later resistance to targeted therapy.

Reviewer #4 (Remarks to the Author):

The authors have addressed all my comments and I recommend publication in Nature Communications.

Reviewer #5 (Remarks to the Author):

The authors have sufficiently addressed my comments and concerns in the revised manuscript. Based on this, I recommend this manuscript be accepted for publication.